# Cerebrospinal fluid-contacting neuron tracing reveals structural and functional connectivity for locomotion in the mouse spinal cord

Yuka Nakamura[1], Miyuki Kurabe[2], Mami Matsumoto[3,4], Tokiharu Sato[1], Satoshi Miyashita[1], Kana Hoshina[1], Yoshinori Kamiya[2], Kazuki Tainaka[1], Hitoshi Matsuzawa[5,6], Nobuhiko Ohno[7,8], Masaki Ueno[1]*

[1]Department of System Pathology for Neurological Disorders, Brain Research Institute, Niigata University, Niigata, Japan; [2]Division of Anesthesiology, Niigata University Graduate School of Medical and Dental Sciences, Niigata, Japan; [3]Section of Electron Microscopy, Supportive Center for Brain Research, National Institute for Physiological Sciences, Okazaki, Japan; [4]Department of Developmental and Regenerative Neurobiology, Institute of Brain Science, Nagoya City University Graduate School of Medical Sciences, Nagoya, Japan; [5]Center for Advanced Medicine and Clinical Research, Kashiwaba Neurosurgical Hospital, Sapporo, Japan; [6]Center for Integrated Human Brain Science, Niigata University, Niigata, Japan; [7]Department of Anatomy, Division of Histology and Cell Biology, Jichi Medical University, School of Medicine, Shimotsuke, Japan; [8]Division of Ultrastructural Research, National Institute for Physiological Sciences, Okazaki, Japan

*For correspondence:
ms-ueno@bri.niigata-u.ac.jp

**Abstract** Cerebrospinal fluid-contacting neurons (CSF-cNs) are enigmatic mechano- or chemo-sensory cells lying along the central canal of the spinal cord. Recent studies in zebrafish larvae and lampreys have shown that CSF-cNs control postures and movements via spinal connections. However, the structures, connectivity, and functions in mammals remain largely unknown. Here we developed a method to genetically target mouse CSF-cNs that highlighted structural connections and functions. We first found that intracerebroventricular injection of adeno-associated virus with a neuron-specific promoter and *Pkd2l1-Cre* mice specifically labeled CSF-cNs. Single-cell labeling of 71 CSF-cNs revealed rostral axon extensions of over 1800 μm in unmyelinated bundles in the ventral funiculus and terminated on CSF-cNs to form a recurrent circuitry, which was further determined by serial electron microscopy and electrophysiology. CSF-cNs were also found to connect with axial motor neurons and premotor interneurons around the central canal and within the axon bundles. Chemogenetic CSF-cNs inactivation reduced speed and step frequency during treadmill locomotion. Our data revealed the basic structures and connections of mouse CSF-cNs to control spinal motor circuits for proper locomotion. The versatile methods developed in this study will contribute to further understanding of CSF-cN functions in mammals.

## Editor's evaluation

This study highlights a novel elegant approach to effectively target neurons contacting the cerebrospinal fluid in mice. The demonstration of the connectivity and roles of CSF-cNs to enhance locomotor speed is convincing and corroborates previous results obtained in mice and zebrafish. Overall this investigation is highly relevant for the sensorimotor and cerebrospinal fluid fields.

## Introduction

Cerebrospinal fluid-contacting neurons (CSF-cNs) are functionally enigmatic cells located in the subependymal area of the central canal in the spinal cord and caudal medulla in vertebrates (*Orts-Del'Immagine and Wyart, 2017*). They have bulbous dendritic structures containing motile cilium and microvilli that extend apically into the CSF. The broad conservation of CSF-cNs in over 200 vertebrate species implicates their fundamental physiological functions (*Agduhr, 1922*; *Agduhr, 1932*; *Kolmer, 1921*). Although the unique structure of CSF-cNs suggests that they are sensory or neurosecretory cells that interact with CSF (*Vígh et al., 2004*; *Vigh and Vigh-Teichmann, 1998*), their specific functions have long remained undetermined.

CSF-cNs have been shown to comprise chemo- and mechanosensory properties that enable the detection of physiological changes in the CSF. The polycystic kidney disease 2-like 1 channel (PKD2L1), a member of the polycystin subgroup of transient receptor potential channels, was originally found to be expressed in the CSF-cNs of mice, and thereafter in other species including zebrafish, rabbits, and macaque monkeys (*Djenoune et al., 2014*; *Tonelli Gombalová et al., 2020*; *Huang et al., 2006*). PKD2L1 is a chemo- or mechanosensitive nonselective cation channel that can sense extracellular pH, osmolarity, and mechanical flows (*Delmas, 2004*; *Orts-Del'immagine et al., 2012*). Indeed, CSF-cNs physiologically respond to alkalinization and acidification in rodents and lampreys, in which acid-sensing ion channel ASICs also contribute to lowered pH (*Huang et al., 2006*; *Jalalvand et al., 2016a*; *Marichal et al., 2009*; *Orts-Del'Immagine et al., 2016*; *Orts-Del'immagine et al., 2012*). CSF-cNs also respond to osmolarity and ATP in rodents and fluid movements in lampreys (*Jalalvand et al., 2016b*; *Marichal et al., 2009*; *Orts-Del'immagine et al., 2012*). These reports indicate that CSF-cNs are chemo- and mechanosensory cells that detect components in the CSF. CSF-cNs have also been reported to be GABAergic interneurons that express glutamate decarboxylase (GAD) and GABA (*Barber et al., 1982*; *Djenoune et al., 2014*; *Stoeckel et al., 2003*). These suggest that componential information of CSF is transferred as inhibitory GABAergic signals onto local circuits.

A series of recent elegant studies in zebrafish larvae have further revealed the connectivity and functions of CSF-cNs. They showed that CSF-cNs are mechanosensory neurons that detect intraspinal mechanical signals to control body posture and locomotion in zebrafish (*Böhm et al., 2016*; *Hubbard et al., 2016*; *Wyart et al., 2009*). In particular, CSF-cNs detect spinal bending through PKD2L1 and control tail beating locomotion (*Böhm et al., 2016*). They send GABAergic inhibitory signals onto glutamatergic V0-v (premotor descending), commissural primary ascending (CoPA) interneurons, and caudal primary (CaP) axial motor neurons (MNs) in local motor circuits to modulate locomotion and control balance during slow and fast swimming (*Fidelin et al., 2015*; *Hubbard et al., 2016*). CSF-cNs further contribute to the maintenance of spine curvature by sensing contacts with the Reissner fiber in the central canal (*Cantaut-Belarif et al., 2018*; *Orts-Del'Immagine et al., 2020*; *Sternberg et al., 2018*; *Troutwine et al., 2020*). These reports indicate that CSF-cNs are intraspinal mechanosensory neurons that control body axis and locomotion. These properties are partially observed in lampreys, in which CSF-cNs respond to spinal bending and pH alteration, leading to a decrease in locomotor burst in an isolated spinal cord (*Jalalvand et al., 2016a*; *Jalalvand et al., 2016b*).

In contrast, the physiological functions and underlying circuitry of CSF-cNs in mammals have been largely undetermined. Although mice are suitable to study connectivity and functions of specific neurons, methods for targeting CSF-cNs are still immature, presumably because their longitudinal and intraspinal location along the central canal make it challenging to access CSF-cNs. The discovery of PKD2L1 expressions in CSF-cNs has allowed to exploit the use of *Pkd2l1-Cre* mice (*Djenoune et al., 2014*; *Huang et al., 2006*; *Jurčić et al., 2021*; *Orts-Del'Immagine et al., 2014*). Fluorescent labeling combined with reporter mice has proved to be a practical method to analyze their morphology and electrophysiological properties, showing that labeled PKD2L1[+] cells have ventral and rostrocaudal axon extensions, respond to pH and osmotic changes, and receive GABA/glycine and glutamatergic inputs (*Jurcic et al., 2019*; *Jurčić et al., 2021*; *Orts-Del'Immagine et al., 2016*; *Orts-Del'immagine et al., 2012*). However, a relatively lower specificity due to ectopic Cre expression in the medullar and spinal parenchyma, as well as extra-brain regions (e.g. the taste buds), makes them insufficient for specific targeting (*Huang et al., 2006*; *Orts-Del'immagine et al., 2012*). The limited specificity, flexibility, and accessibility of current methods in mice are a barrier to investigate their structure, network, and function. Although a very recent study used *Pkd2l1[IRES-Cre]* and reporter mice to show local connections of CSF-cNs with spinal neurons and their contribution to skilled locomotion (*Gerstmann et al.,*

*2022*), alternative specific and flexible methods for manipulating CSF-cNs are required to determine precise features of structure, connectivity, and functions.

In this study, we serendipitously discovered a method to target mouse CSF-cNs via intracerebroventricular injection of adeno-associated virus (AAV) with a neuron-specific promoter, which enabled us to introduce any genes into the CSF-cNs. We developed versatile methods to label and manipulate CSF-cNs, and these facilitated to elucidate the precise and detailed structure, connectivity, and function of mouse CSF-cNs in the spinal motor circuitry that are involved in locomotor control. The methodological tools and approach described herein will provide further insight to understand the enigmatic functions of CSF-cNs in mammals.

## Results

### Intracerebroventricular AAV injection specifically labels CSF-CNs

In the course of our previous studies to label projections of the corticospinal tract that extend from the cerebral cortex to spinal cord (*Ueno et al., 2018*), we serendipitously found that injection of a fluorescent protein-expressing AAV into the sensorimotor cortex occasionally labeled cells around the central canal of the spinal cord (*Figure 1A and B*). We presumed that AAV particles may have leaked into the lateral ventricle, which is located just beneath the cortex and passed through the CSF to label the cells in the spinal cord. To examine this possibility, we directly injected AAV that expressed mCherry under the human synapsin I promoter (AAV-syn-mCherry, serotype 1/2) into the lateral ventricle (*Figure 1A*). We found that a direct intracerebroventricular injection efficiently labeled the cells surrounding the central canal (*Figure 1C*). The cells were mostly located at the ventral position of the central canal underlying the ependymal cells and had dendrite-like structures extending directly into the central canal (*Figure 1D*). Their location and structure likely corresponded to those of CSF-cNs, which are present in mice (*Djenoune et al., 2014*). We found that the AAV-labeled cells mostly co-expressed PK2DL1, a specific marker of CSF-cNs, throughout the rostrocaudal axis of the spinal cord (*Figure 1D*, *Figure 1—figure supplement 1A-D*; PK2DL1$^+$/mCherry$^+$: cervical, 99.2 ± 0.49%; thoracic, 93.4 ± 0.67%; lumbar, 97.3 ± 0.61%; n=4) (*Huang et al., 2006*). This demonstrates that an intracerebroventricular injection of AAV specifically labels CSF-cNs in the spinal cord. Although PKD2L1$^+$ CSF-cNs are located in the dorsolateral and ventral portions of the central canal (*Orts-Del'Immagine et al., 2014*), AAV mostly labeled the ventral cell population, rather than the dorsolateral ones (cervical, 80.0 ± 5.50%; thoracic, 95.3 ± 0.38%; lumbar, 88.5 ± 2.41%; n=4; *Figure 1D*). We further examined methodological factors required for specific labeling. Instead of the neuron-specific promoter, we tested a pan-cellular CAG promoter. However, the use of AAV with the CAG promoter labeled not only CSF-cNs but also ependymal and meningeal cells (*Figure 1—figure supplement 1E–H*). This indicates that a neuron-specific promoter is required for specific CSF-cN labeling.

This method enabled us to investigate the morphology of CSF-cNs in detail. We first observed that, in transverse sections, the cells projected fibers into the restricted area of the ventral funiculus and formed large bundles (*Figure 1C*, *Figure 1—figure supplement 1A-D*). The location of the bundles was gradually shifted, depending on the spinal level. In the cervical and rostral thoracic cord, the bundles formed several clusters in the dorsal area of the ventral funiculus (*Figure 1C*, *Figure 1—figure supplement 1A, B*). In contrast, in the lumbar and sacral cord, the bundles formed a single large bundle that was located more medioventrally, aligning with the ventral medial fissure (*Figure 1—figure supplement 1C, D*). The axons in the bundles exhibited a bouton-like structures in transverse sections. We stained the sections with presynaptic markers of inhibitory neurons: GAD65 (GAD2), GAD67 (GAD1), and VGAT. They showed a unique staining accumulated in the ventral mCherry$^+$ fiber clusters (*Figure 1E*), which was consistent with the GABAergic properties of CSF-cNs (*Djenoune et al., 2014*; *Stoeckel et al., 2003*). These results indicate that CSF-cNs form fiber bundles containing GABA-related proteins in the ventral funiculus.

To delineate the structures of CSF-cNs, we further examined the morphology of mCherry$^+$ cells in thick horizontal sections. In the subependymal area, the cells are mutually aligned on each side of the spinal cord, and extended dendritic bulbs that were precisely aligned and occupied most of the ventral surface of the central canal (*Figure 1F*). At the ventral level, we found that mCherry$^+$ fibers extended rostrocaudally with GAD65$^+$ puncta, which corresponded to the fiber bundles seen in the transverse sections (*Figure 1E and G*). We further identified unique rostrocaudally projecting

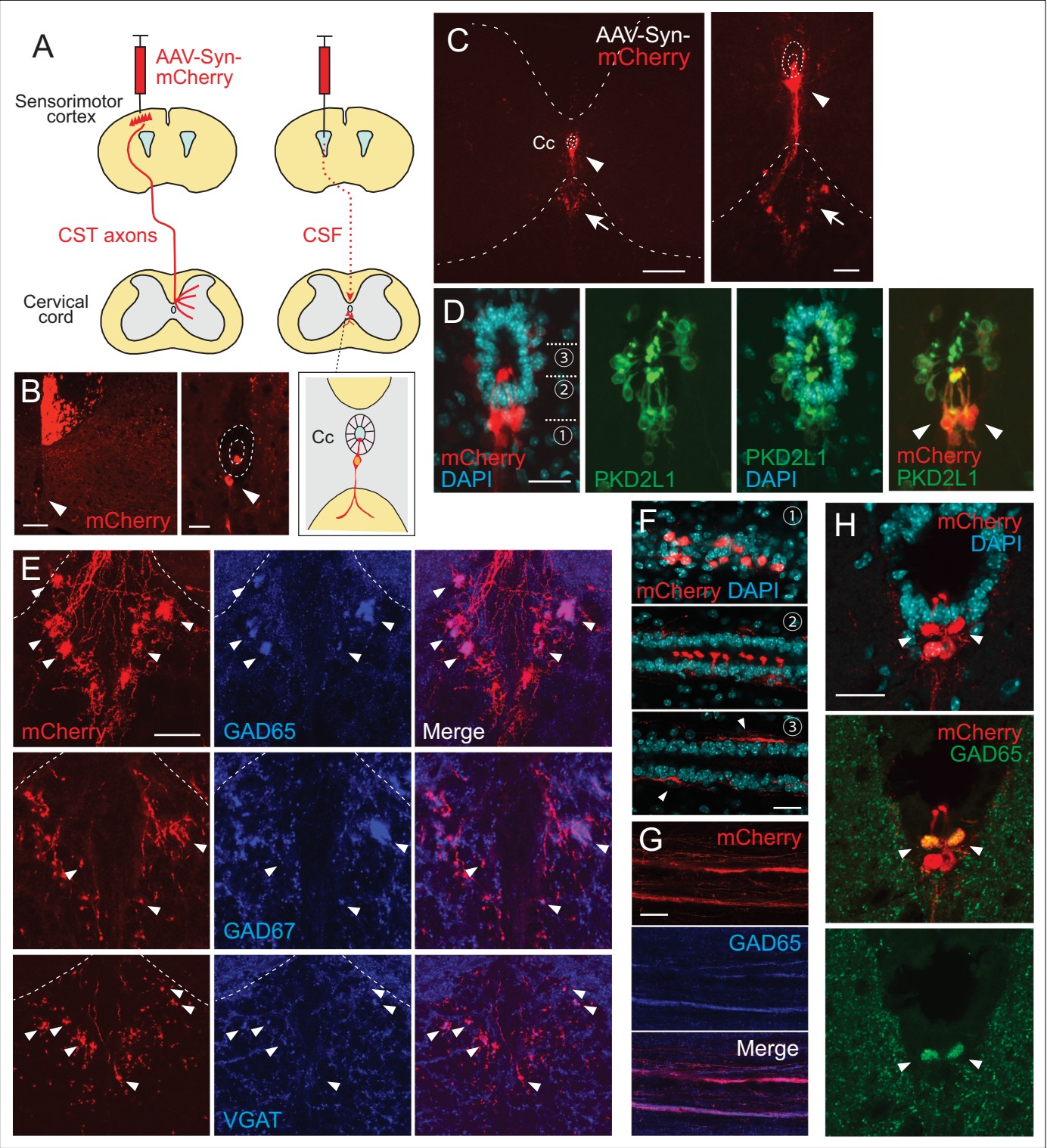

**Figure 1.** Intracerebroventricular injection of adeno-associated virus (AAV) with a neuron-specific promoter specifically labels cerebrospinal fluid-contacting neurons (CSF-cNs). (**A**) Schema of AAV-Syn-mCherry injections into the sensorimotor cortex (left) or lateral ventricle (right). (**B**) mCherry-labeled neurons around the central canal (Cc) occasionally observed after cortical AAV injections, which were putatively labeled by AAV leaking into CSF (red, arrowhead). The right panel is a magnified view around the Cc (dotted lines). Note that abundant corticospinal tract (CST) axons are also labeled. (**C**) Representative transverse images of mCherry-labeled neurons around the Cc at the cervical level after an AAV injection into the lateral ventricle (red, arrowheads). The right panel is a magnified view around the Cc. Arrows represent mCherry$^+$ fibers extending to the ventral funiculus. Dotted lines indicate Cc and the border between the gray and white matter. (**D**) Confocal views of mCherry-labeled neurons (red) around the Cc, co-

*Figure 1 continued on next page*

*Figure 1 continued*

expressing PKD2L1 (green), a marker of CSF-cNs (arrowheads). 4',6-diamidino-2-phenylindole (DAPI), blue. (**E**) mCherry$^+$ axon bundles extending into the ventral funiculus (red) with GAD65, GAD67, and VGAT signals (blue, arrowheads). Dotted lines indicate the border between the gray and white matter. (**F**) Horizontal views of mCherry$^+$ CSF-cNs (red) in different planes of depths (1–3), which are indicated in (**D**). Confocal analyses. Arrowheads, subependymal fibers. (**G**) Horizontal views of mCherry$^+$ axon bundles in the ventral funiculus (red), co-expressing GAD65 (blue). (**H**) Transverse images of mCherry$^+$ axon bundles in the subependymal area, co-labeled with GAD65 (green, arrowheads). Scale bars, 100 µm (left panel of **B**); 25 µm (right panel of **B**, **D**, **F**); 200 µm (left panel of **C**); 50 µm (right panel of **C**, **E**, **H**).

The online version of this article includes the following figure supplement(s) for figure 1:

**Figure supplement 1.** Specific labeling of cerebrospinal fluid-contacting neurons (CSF-cNs) by intracerebroventricular injections of adeno-associated virus (AAV).

mCherry$^+$ fiber bundles located just beneath the ependymal cells (*Figure 1F*, arrowheads). These subependymal bundles were also labeled by GAD65 and were primarily observed at the cervical cord, but not at other spinal levels (*Figure 1H*).

## Single-cell sparse labeling in a cleared tissue reveals axon trajectories and recurrent connections of CSF-CNs

Leveraging the labeling method together with a tissue clearing technique, CUBIC (clear, unobstructed brain/body imaging cocktails and computational analysis; *Tainaka et al., 2018*), we further analyzed the three-dimensional (3D) structure of CSF-cNs. Transparent spinal cords clearly exhibited mCherry$^+$ cells that were aligned along the central canal, and their fibers extended ventrally and then rostrocaudally with forming bundles in the ventral funiculus (*Figure 2A–C*, *Figure 2—figure supplement 1I-L*; *Video 1*).

We next attempted to sparsely label the cells to trace the structures of CSF-cNs at a single-cell level. A low dose of AAV-Syn-Cre was intraventricularly injected into *Rosa26*$^{CAG-lox-stop-lox-tdTomato}$ mice (*Rosa26*$^{lsl-tdTomato}$; Ai14), in which a low probability of Cre expression drives sufficient amount of fluorescent proteins (*Winnubst et al., 2019*). tdTomato signals which were enhanced by immunostaining could delineate the morphology of a single CSF-cN (*Figure 2E*), although it also increased tdTomato$^+$ signals in the vasculature presumptively leaked in Ai14 line. On the apical side, tdTomato$^+$ dendritic bulbs protruded to the CSF and occasionally contained cilia-like structures. The soma had multiple short dendritic neurites on the basal side (*Figure 2E*; *Video 2*).

Notably, the single-cell tracing determined the patterns of axon trajectories. We succeeded in tracing 71 cells (cervical, n=25; thoracic, n=28; lumbar, n=10; and sacral, n=8), and additional 261 cells that were traced partially up to the ventral funiculus (cervical, n=62; thoracic n=81; lumbar, n=85; and sacral, n=33). Most of the cells had a single elongated axon that first projects ventrally and then rostrally in the ventral funiculus (*Figure 2E–G*, *Figure 2—figure supplement 1*; *Video 3*). All of them projected to specific side of the ventral funiculus without bilateral projections (left side, 46.5%; right side, 53.5%), although it was unclear if they extended to ipsi- or contralateral side because their soma were located at the ventral midline along the central canal (67/71 cells). The remaining four dorsolateral cells projected axons ipsilaterally. A small proportion of the cells at the cervical and sacral levels also exhibited caudal projections in the ventral funiculus (4.2% in total) or bidirectional projections at the cervical level (1.2%; two cells at C2 and C4/5, respectively, with three and one cells located ventral and lateral to the central canal; *Figure 2H*, *Figure 2—figure supplement 1F*). Most of the fibers extended rostrally to an average length of approximately 1800–4800 µm and tended to be longer in CSF-cNs located in the lumbar spinal cord (*Figure 2I*). They often formed parallel collaterals that extended rostrally in the ventral funiculus (*Figure 2I*), wherein none of them crossed the midline. We further found that rostrally extending fibers had several dorsal collaterals that turned back to the central canal (*Figure 2E–G*, *Figure 2—figure supplement 1A-E*; *Video 3*). The fibers terminated in the region surrounding the central canal with abundant branching fibers, extending to the ipsi- and contralateral sides in substantial population (ipsilateral, 7.0%; bidirectional, 62.8%; contralateral, 9.3%; midline beneath the central canal, 20.9%). These collaterals were more abundant in cells of the cervical cord than in those of the caudal spinal cord (*Figure 2I*, *Figure 1—figure supplement 1I-L*). Some of the fibers further extended along the basal side of ependymal cells (*Figure 2E and F*, *Figure 2—figure supplement 1A*), some of which possibly corresponded to the bundles observed in the subependymal area (*Figure 1F and H*).

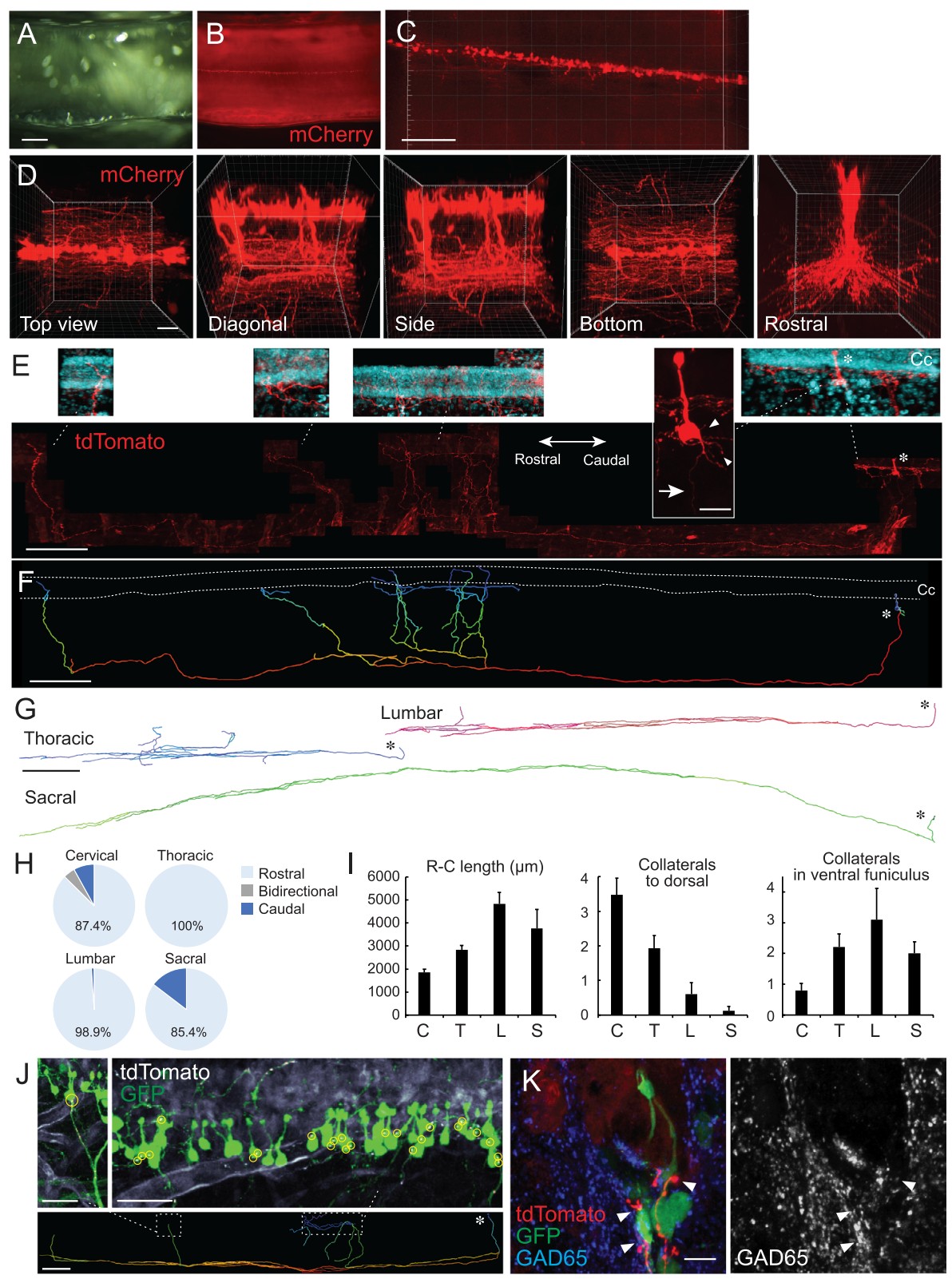

**Figure 2.** Single-cell tracing in a cleared spinal cord reveals structures and projection patterns of cerebrospinal fluid-contacting neurons (CSF-cNs). (**A, B**) Clearing of adeno-associated virus (AAV)-Syn-mCherry-injected cervical cord with CUBIC. Cleared cervical cord (**A**) and a fluorescent image (**B**) shown in a side view under a stereomicroscope. (**C, D**) Z-stack confocal images of mCherry⁺ CSF-cNs (**C**), red, and higher magnification views from the top, diagonal, side, bottom, and rostral sides (**D**), representing aligned soma and their ventral to rostrocaudal axon projections. (**E**) Representative images

*Figure 2 continued on next page*

*Figure 2 continued*

of a single-cell traced tdTomato⁺ CSF-cN and its projections (red). A side view at the cervical level of *Rosa26ˡˢˡ⁻ᵗᵈᵀᵒᵐᵃᵗᵒ* mice injected with a low dose of AAV-Syn-Cre. Top panels depict higher magnification views of the tdTomato⁺ cell and its projections around the central canal (Cc). 4',6-diamidino-2-phenylindole (DAPI; blue). Arrow, axon; arrowheads, basal dendrites; asterisks, cell bodies. Left, rostral; right, caudal. (**F**) A reconstructed IMARIS image of the single-cell traced CSF-cN in (**E**). Dotted lines, Cc; asterisk, cell body. (**G**) Representative reconstructed images of single-cell traced CSF-cNs in the thoracic, lumbar, and sacral levels. Asterisks, cell bodies. (**H**) Pie charts representing the ratio of CSF-cNs comprising rostral, caudal, or bidirectional axon projections (cervical, n=87; thoracic, n=109; lumbar, n=95; sacral, n=41). (**I**) Rostrocaudal axonal lengths in the ventral funiculus and the number of collaterals projecting in dorsal and rostral directions in the ventral funiculus. Cervical (**C**), n=25; thoracic (**T**), n=28; lumbar (**L**), n=10; sacral (**S**), n=8; the mean ± standard error of the mean (SEM). (**J**) tdTomato⁺ axons of single-cell traced CSF-cNs (white) contacting onto other GFP⁺ CSF-cNs (green), labeled with a high-dose of AAV-Syn-EGFP injection. Contact areas are indicated in yellow circles. Images in the upper panels are of the dotted areas in the bottom IMARIS-reconstructed image. (**K**) tdTomato⁺ axons of single-cell traced CSF-cNs (red) contacting the soma of other GFP⁺ CSF-cNs (green) with a presynaptic marker GAD65 (blue, left; white, right; arrowheads). Scale bars, 500 μm (**A, B**); 200 μm (**C**), lower panel of (**E**), (**F**), lower panel of (**J**); 50 μm (**D**); 20 μm (upper panel of **E**); 500 μm (**G**); 50 μm (upper panels of **J**); 10 μm (**K**).

The online version of this article includes the following source data and figure supplement(s) for figure 2:

**Source data 1.** Raw data for direction, length, and collaterals of CSF-cN projections.

**Figure supplement 1.** Projections of single-cell traced cerebrospinal fluid-contacting neurons (CSF-cNs).

The fibers that turned back to the subependymal area may have contacted other cells surrounding the central canal, such as CSF-cNs. To clarify possible CSF-cN–CSF-cN connections, we made transparent spinal cord samples containing a single-cell traced tdTomato⁺ axons and GFP-labeled CSF-cNs by simultaneously injecting a low dose of AAV-Syn-Cre and a higher amount of AAV-Syn-EGFP into the ventricle. We found that the cell bodies of GFP⁺ CSF-cNs frequently received contacts from tdTomato⁺ fibers that ascended dorsally (*Figure 2J*). In some cases, a single tdTomato⁺ axon was connected to multiple CSF-cNs, particularly at the basal or apical side of the cell bodies (*Figure 2J*). We further made transverse sections from the cleared spinal cord, and confocal microscopic analyses showed that the GFP⁺ CSF-cN soma received multiple contacts from tdTomato⁺ fibers with GAD65⁺ presynaptic boutons (*Figure 2K*; *Video 4*). These observations indicate that CSF-cNs project rostrally and form recurrent loop connections with rostrally located CSF-cNs.

## Combining *Pkd2l1-Cre* mice improves specificity of CSF-CNs labeling

Although the developed method was efficient for the labeling of CSF-cNs, other cells in the supraspinal brain regions were also possibly labeled. This may limit the applicability of this approach for the experiments which require a higher level of specificity such as in ultrastructural and functional analyses. Indeed, we found that the intracerebroventricular injection of AAV-Syn-mCherry also labeled cells surrounding the ventricle in the brain, especially those in the circumventricular organs including the subfornical organ, vascular organ of the lamina terminalis, and area postrema, in which PKD2L1

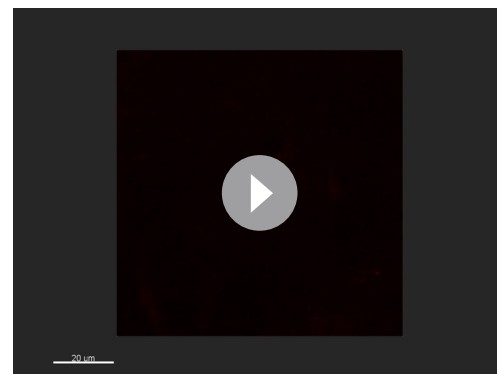

**Video 1.** Magnified z-stack images of mCherry-labeled cerebrospinal fluid-contacting neurons in a cleared cervical cord.

https://elifesciences.org/articles/83108/figures#video1

**Video 2.** Z-stack images of a tdTomato⁺ cell body of single-cell traced cerebrospinal fluid-contacting neuron.

https://elifesciences.org/articles/83108/figures#video2

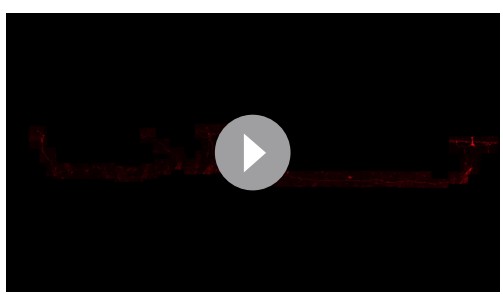

**Video 3.** Z-stack images of tdTomato⁺ projections of single-cell traced cerebrospinal fluid-contacting neuron in a cleared cervical cord.
https://elifesciences.org/articles/83108/figures#video3

was not expressed (*Figure 3A–D*, *Figure 3—figure supplement 1A–D*). Cells aligned with the lateral and third ventricles and cerebral aqueduct, as well as a few neurons in the olfactory bulb and cerebellum, were also ectopically labeled (*Figure 3A, C and M*, *Figure 3—figure supplement 1A, C, D*). A very small number of cortical neurons at the injection site were also occasionally labeled.

To increase the specificity, we tested to inject AAV containing a Cre-dependent expression construct DIO (or FLEX) into *Pkd2l1-Cre* mice (*Huang et al., 2006*). We first validated Cre expressions in mice crossed with the CAG-lox-CAT-lox-EGFP (CAG-lcl-EGFP) reporter line. Abundant numbers of EGFP⁺ CSF-cNs were found around the central canal; however, we noticed that other cells also expressed GFP in the gray matter in adult mice (*Figure 3E*), which were not characterized previously (*Orts-Del'immagine et al., 2012*) and in another *Pkd2l1^IRES-Cre* line (*Gerstmann et al., 2022*). They had glial morphologies and were colocalized with an oligodendrocyte marker Olig2 (*Figure 3F*) but not with the astrocyte marker Sox9 (data not shown). We reassessed previously published single-cell/nucleus RNA sequencing data (*Russ et al., 2021*), and it showed that a subpopulation of oligodendrocytes actually expressed *Pkd2l1* mRNA (*Figure 3G*). In contrast, we did not observe any *Pkd2l1*-EGFP⁺ cells surrounding the ventricles in the brain. We injected AAV-Syn-DIO-hM4Di-mCherry intracerebroventricularly into *Pkd2l1-Cre* mice, and it specifically labeled CSF-cNs in the spinal cord without any expressions throughout the brain (*Figure 3H–M*, *Figure 3—figure supplement 1E-H*; PK2DL1⁺/mCherry⁺: cervical, 100 ± 0%; thoracic, 100 ± 0%; lumbar, 99.3 ± 0.65%; n=3). Thus, the combined use of *Pkd2l1-Cre* mice and AAV-DIO improved specificity, which would be useful for experiments requiring the strict targeting of CSF-cNs.

## Serial block-face scanning electron microscopy (SBF-SEM) with specific peroxidase-based labeling reveals ultrastructure and recurrent connections of CSF-CNs

Previous studies have reported ultrastructural features of rodent CSF-cNs using electron microscopy (*Bjugn et al., 1988*; *Tonelli Gombalová et al., 2020*; *Marichal et al., 2009*; *Stoeckel et al., 2003*). In order to unequivocally identify and clarify the structures and connectivity of CSF-cNs in the spinal cord, we performed serial block-face scanning electron microscopy (SBF-SEM) analyses (*Nguyen et al., 2016*; *Thai et al., 2016*), combined with AAV-mediated COX4-dAPEX2 and synaptophysin (SYP)-HRP labeling, which tag mitochondria and synaptic vesicles by peroxidase-reactive 3, 3'-diaminobenzidine (DAB) staining, respectively (*Zhang et al., 2019*). To avoid any leaks at the ultrastructural level, AAV-Syn-DIO-COX4-dAPEX2 or SYP-HRP was injected into *Pkd2l1-Cre* mice to label CSF-cNs.

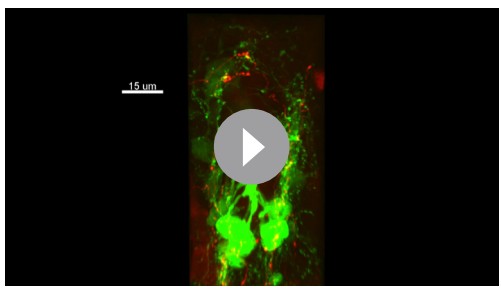

**Video 4.** Z-stack images of tdTomato⁺ projections of single-cell traced cerebrospinal fluid-contacting neurons (CSF-cNs) innervating to GFP⁺ CSF-cNs in a cleared cervical cord.
https://elifesciences.org/articles/83108/figures#video4

Under light microscopy, it was observed that COX4-dAPEX2 labeled soma and ventral axon bundles (*Figure 4A*, *Figure 4—figure supplement 1A*). SYP-HRP labeled dendritic bulbs, ventral axon bundles, and presumptive dorsal projections, while the signals were weak in the soma (*Figure 4B*, *Figure 4—figure supplement 1B*).

In SBF-SEM, DAB⁺ COX4-dAPEX2 and SYP-HRP signals were detected in the mitochondria and vesicles of CSF-cNs, respectively, in the subependymal area (*Figure 4C–I*). Dendritic bulbs contained a rich amount of DAB⁺ mitochondria and vesicles (*Figure 4D, I*, *Figure 4—figure supplement 1K*). At the soma, axon terminals

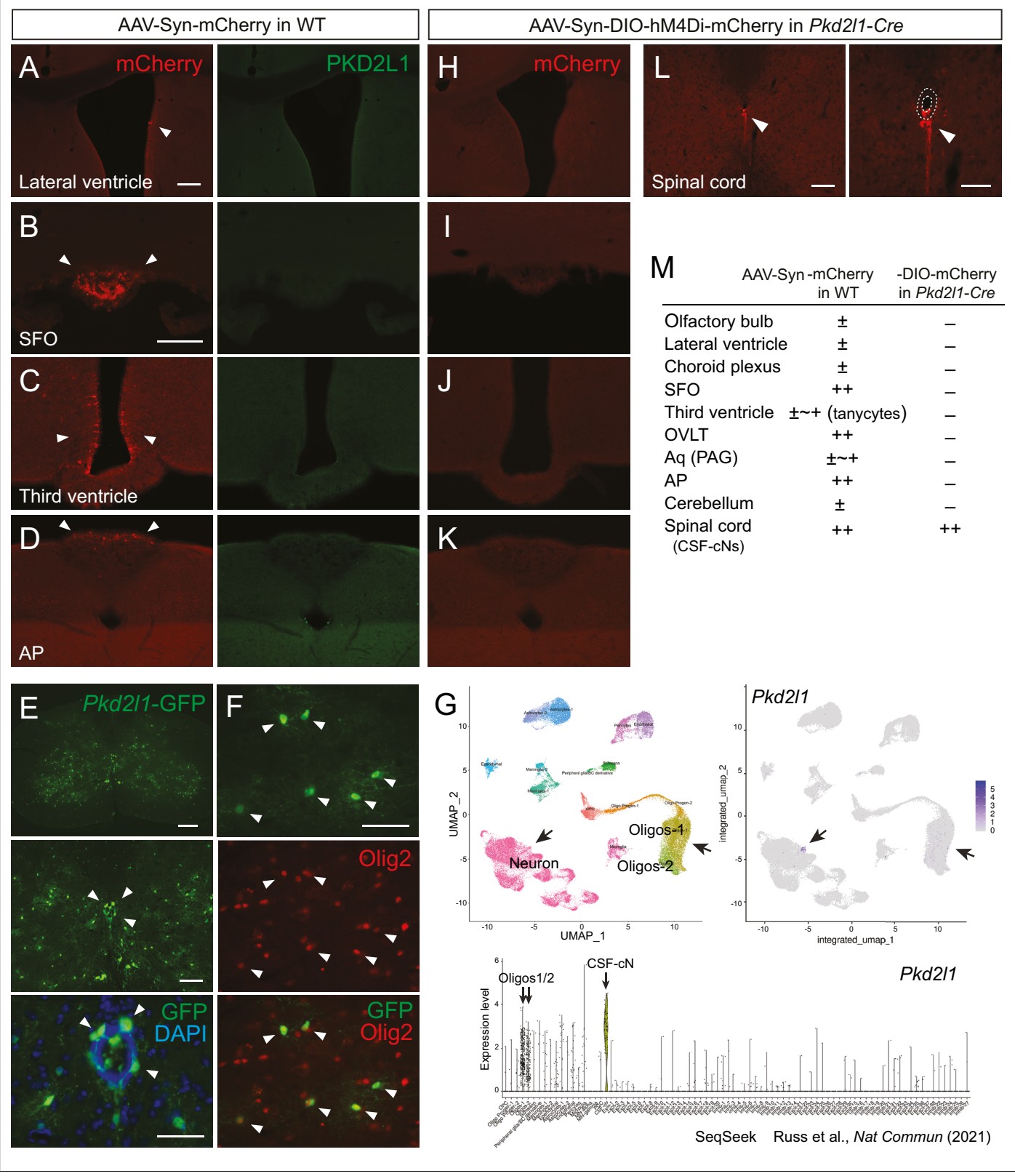

**Figure 3.** Injection of adeno-associated virus (AAV)-Syn-DIO construct into *Pkd2l1-Cre* mice increases labeling specificity of cerebrospinal fluid-contacting neurons (CSF-cNs). (**A–D**) mCherry labeled cells (red, arrowheads) in the lateral ventricle (**A**), SFO (**B**), third ventricle (**C**), and AP (**D**) following AAV-Syn-mCherry intracerebroventricular injection into wild-type (WT) mice. No PKD2L1 expression (green) was observed in these regions. (**E, F**) GFP⁺ signals in the CSF-cNs (**E**), green, arrowheads and Olig2⁺ oligodendrocytes (**F**), red, arrowheads in the spinal cord of an adult *Pkd2l1-Cre*;CAG-lcl-

*Figure 3 continued on next page*

*Figure 3 continued*

EGFP mouse. 4',6-diamidino-2-phenylindole (DAPI), blue. (**G**) Single-cell/nucleus RNA-seq data in SeqSeek database (https://seqseek.ninds.nih.gov/home, *Russ et al., 2021*) represent *Pkd2l1* mRNA expression in CSF-cNs and a subpopulation of oligodendrocytes. Upper panels show a uniform manifold approximation and projection (UMAP) plot of spinal cord single-cell/nucleus RNA-seq data (left), and feature plots highlight *Pkd2l1* expression in spinal cells (violet, right). The bottom panel depicts violin plots representing *Pkd2l1* expression levels in spinal cell types. (**H–K**) mCherry-labeled cells (red, arrowheads) in the lateral ventricle (**H**), SFO (**I**), third ventricle (**J**), AP (**K**), and cervical cord (**L**) following AAV-Syn-DIO-hM4Di-mCherry intracerebroventricular injection into *Pkd2l1-Cre* mice. The right image of (**L**) is a higher magnification view around the central canal (dotted lines). (**M**) Summary of AAV-Syn-mCherry and -DIO-hM4Di-mCherry-labeled cells in the brain and spinal cord of WT and *Pkd2l1-Cre* mice, respectively. Scale bars, 100 µm (**A, B**); 200, 100, 50 µm (upper to bottom in **E**); 50 µm (**F**); 100, 50 µm (left, right in **L**). AP, area postrema; Aq, cerebral aqueduct; OVLT, vascular organ of the lamina terminalis; PAG, periaqueductal gray; SFO, subfornical organ.

The online version of this article includes the following source data and figure supplement(s) for figure 3:

**Source data 1.** Summary of adeno-associated virus-Syn-mCherry and -DIO-hM4Di-mCherry-labeled cells in the brain and spinal cord of wild-type and *Pkd2l1-Cre* mice (related to *Figure 3M*).

**Figure supplement 1.** Adeno-associated virus (AAV)-Syn-DIO injections into *Pkd2l1-Cre* mice do not label ventricular cells in the brain.

that contained dense synaptic vesicles were frequently apposed, indicating the presence of synaptic inputs. Interestingly, we found that some of them contained COX4-dAPEX2+ mitochondria or SYP-HRP+ vesicles (*Figure 4F, G–I*, *Figure 4—figure supplement 1C-E*). We reconstructed the 3D structure of SYP-HRP+ CSF-cNs from serial section images (*Figure 4J*, *Video 5*). It delineated apical dendritic bulbs, short dendritic structures, and basal axons, and further revealed that CSF-cNs typically received SYP-HRP+ presynaptic contacts onto the apical and basal parts of the soma (*Figure 4G–J*, *Figure 4—figure supplement 1D-E*). We further performed transmission electron microscope (TEM) analyses, which clearly showed that presynaptic terminals containing COX4-dAPEX2+ mitochondria and abundant synaptic vesicles contacted COX4-dAPEX2+ CSF-cNs (*Figure 4—figure supplement 1G, H*). These results strongly support the observation in a single-cell tracing that showed a recurrent network of CSF-cNs (*Figure 2*).

The SYP-HRP and COX4-dAPEX2 labeling also enabled to track axon trajectories. We found a unique prominent bundle structures composed of a number of non-myelinated axons in the subependymal area and ventral funiculus, in which the axons frequently contained COX4-dAPEX2+ mitochondria or SYP-HRP+ vesicles (*Figure 4E, G–I*, *Figure 4—figure supplement 1F*). TEM analyses also clearly showed COX4-dAPEX2+ mitochondria in these unmyelinated axons (*Figure 4—figure supplement 1I, J*), which corresponded to axon bundles of CSF-cNs (*Figure 1E and H*). Taken together, serial electron microscopic analyses with CSF-cN labeling revealed the ultrastructure of the soma and axons and identified the recurrent connections of CSF-cNs.

## Functional responses of recurrent connections between CSF-cNs

We next investigated functional connections between CSF-cNs by using electrophysiological recordings combined with optogenetic stimulation. We injected AAV-Syn-DIO-channelrhodopsin-2 (ChR2)-mCherry at T9 level of *Pkd2l1-Cre;CAG-lcl-EGFP* mice to express ChR2 in local CSF-cNs and then obtained whole-cell patch clamp recordings from *Pkd2l1*-GFP+ CSF-cNs in rostral thoracic slices at T4–6, in which rostrally extending ChR2-mCherry+ fibers were preserved (355.3±52.5 fibers at the ventral bundle, n=3; *Figure 5A*). A blue light stimulation (470 nm, 10ms) evoked inhibitory postsynaptic currents (IPSCs) in 10 of 12 cells with an average latency of 9.47±0.66 ms and time to peak of 5.87±1.15 ms (*Figure 5B*). The responses were blocked following the treatment with bicuculline, a GABA$_A$ receptor antagonist (8/8 cells; *Figure 5C*), indicating that the IPSCs were mediated by GABA$_A$ receptors. No responses were detected in the control spinal slices injected with AAV-Syn-DIO-mCherry (0/14 cells; *Figure 5D*). The electrophysiological data indicate that CSF-cNs functionally inhibit the activities of rostral CSF-cNs.

## Intraspinal connectivity of CSF-CNs with motor-related neurons

Previous studies have shown that CSF-cNs in zebrafish larvae are connected with premotor and sensory interneurons (V0-v, CoPA) and MNs (CaP) (*Fidelin et al., 2015*; *Hubbard et al., 2016*). This motivated us to investigate their connectivity with motor-related neurons in mice. We first examined connections with Chat+ cholinergic neurons, which include MNs and interneurons mainly located in Rexed lamina VII and X around the central canal (*Barber et al., 1984*; *Miles et al., 2007*; *Zagoraiou*

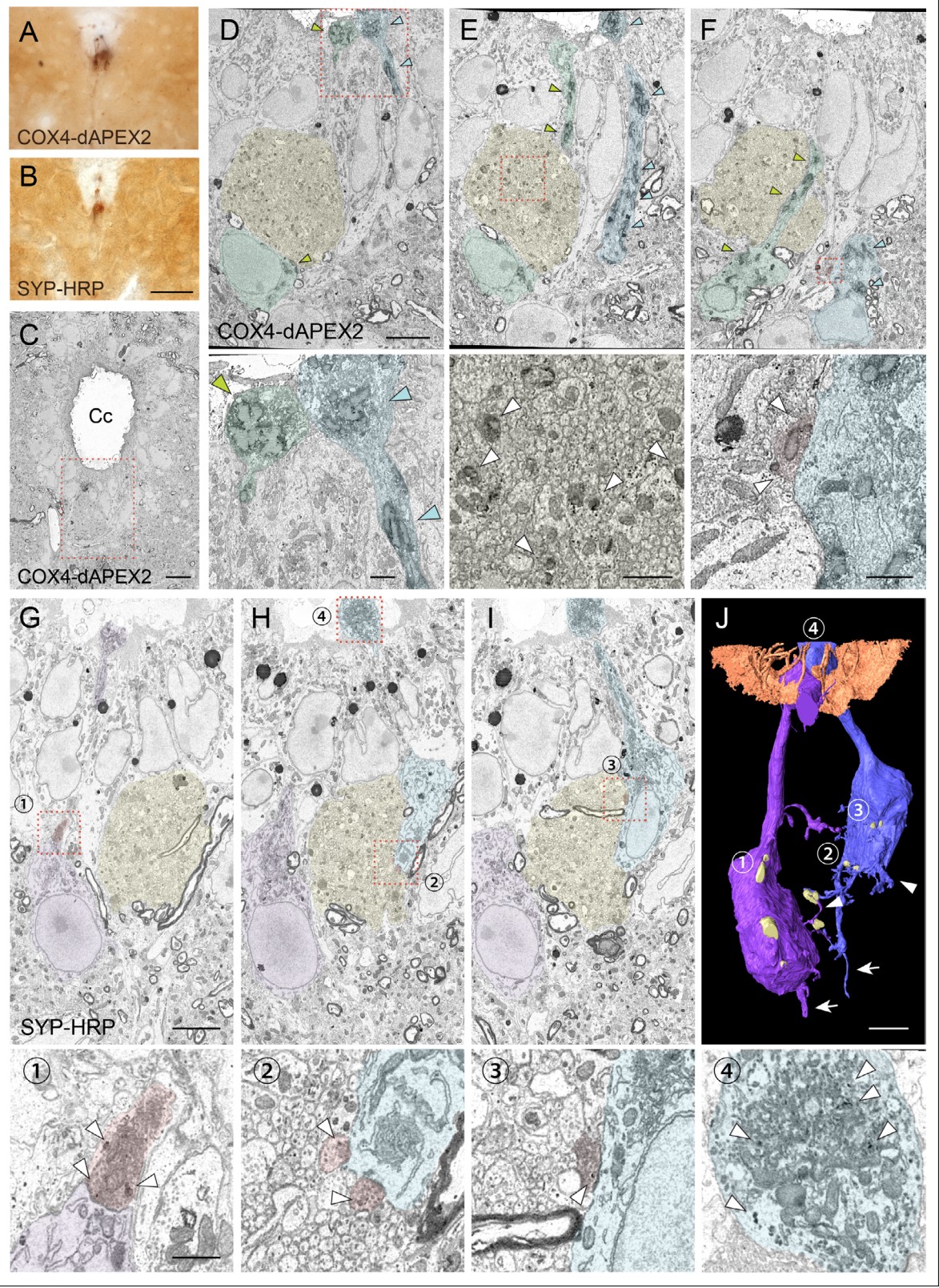

**Figure 4.** Serial block-face scanning electron microscopy (SBF-SEM) analyses with COX4-dAPEX2 and SYP-HRP labeling reveal structures and recurrent connections of cerebrospinal fluid-contacting neurons (CSF-cNs). (**A, B**) Representative light microscopic images of 3, 3'-diaminobenzidine (DAB)-stained spinal cord sections of *Pkd2l1-Cre* mice with injection of adeno-associated virus (AAV)-Syn-DIO-COX4-dAPEX2 (**A**) or AAV-Syn-DIO-SYP-HRP (**B**). (**C**) A low-magnification electron microscopic image of the cervical cord around the central canal (Cc) in *Pkd2l1-Cre* mice injected with AAV-Syn-DIO-

*Figure 4 continued on next page*

*Figure 4 continued*

COX4-dAPEX2. The dotted area was subjected to SBF-SEM analyses and is shown in (**D–F**). (**D–F**) Representative serial (the 130th, 148th, and 186th) SBF-SEM images of the COX4-dAPEX2-labeled sample. High-magnification images of the dotted areas in the upper panels are shown in the bottom panels, which represent dendritic bulbs (**D**), subependymal axon bundles (**E**), and a synaptic contact (**F**) that include DAB-positive mitochondria. COX4-dAPEX2[+] CSF-cNs are pseudo-colored in green and blue; the subependymal unmyelinated axon bundle in yellow; the presynaptic terminal in red. Arrowheads indicate DAB-positive mitochondria labeled with COX4-dAPEX2. (**G–I**) Representative serial SBF-SEM images (the 173th, 224th, and 249th) of the SYP-HRP-labeled sample in the ventral part of Cc in the cervical cord. High-magnification images of the dotted areas in the top panels are shown in the bottoms, which represent synaptic contacts (1–3), and a dendritic bulb (4) containing SYP-HRP[+] vesicles. SYP-HRP[+] CSF-cNs are pseudo-colored in purple and blue; the subependymal axon bundle in yellow; the presynaptic terminals in red. Arrowheads, SYP-HRP[+] vesicles. (**J**) Three-dimensional reconstruction of SYP-HRP[+] CSF-cNs and presynaptic terminals (yellow), represented in (**G–I**). The numbers 1–4 indicate the positions shown in (**G–I**). Microvilli and cilia of ependymal cells are represented in brown to show the surface of the Cc. Arrows, axons; arrowheads, basal dendrites. Scale bars, 50 μm (**A, B**); 10 μm (**C**); 5 μm (upper panels of **D–J**); 1 μm (bottom panels of **D–I**).

The online version of this article includes the following figure supplement(s) for figure 4:

**Figure supplement 1.** Serial block-face scanning electron microscopy (SBF-SEM) and transmission electron microscope (TEM) analyses in the spinal cord of *Pkd2l1-Cre* mice injected with adeno-associated virus (AAV)-Syn-DIO-COX4-dAPEX2 and SYP-HRP.

*et al., 2009*). In cholinergic neuron-specific *Chat^Cre*;CAG-lcl-EGFP mice, abundant *Chat*-GFP[+] fibers elongated from the ventral gray matter to the area around the central canal and ventral funiculus, where AAV-labeled mCherry[+] CSF-cNs and their fibers were located (*Figure 6A–C*). We found that *Chat*-EGFP[+] neurites and cell bodies were frequently contacted by mCherry[+] CSF-cN fibers with the inhibitory postsynaptic marker gephyrin around the central canal and ventral midline (*Figure 6D–H, Figure 6—figure supplement 1A-C*). This suggests that CSF-cNs have anatomical connections with Chat[+] central canal cluster cells, which are located around the central canal, or neurites of V0c neurons (partition cells) located in the medial to lateral intermediate gray matter (*Barber et al., 1984*; *Miles et al., 2007*; *Zagoraiou et al., 2009*). In addition, neurites of medial MNs that likely extend near the central canal may also participate in these connections. We further observed *Chat*-EGFP[+] neurites intermingled into subependymal and ventral mCherry[+] axon bundles (*Figure 6D, I–K*), which may correspond to subependymal longitudinal dendrites of central canal cluster cells and medially extended neurites of medial MNs, respectively (*Barber et al., 1984*; *Phelps et al., 1984*). The *Chat*-EGFP[+] neurites extending to the ventral bundle contained gephyrin[+] puncta, which were contacted by mCherry[+] axons (*Figure 6K*).

Although MNs are located far from the central canal in laminae IX, they have arborized dendrites that widely extend from the soma (*Rotterman et al., 2014*). Since the *Chat*-EGFP[+] neurites might include MN dendrites, we next specifically labeled MNs by injecting retro-AAV-CAG-EGFP into muscles (*Figure 6L*). MN pools of axial trunk and proximal/distal limb muscles are located in the ventromedial and ventrolateral gray matter, respectively, and have distinct areas of dendrite innervations (*Goetz et al., 2015*). In accordance, EGFP[+] MN somas of axial muscles (neck and back) were mostly located ventromedially (24/29 cells for the neck and 26/26 cells for the back muscle), and dendritic fibers elongated to the area around the central canal (*Figure 6M, Figure 6—figure supplement 1D, E*). In contrast, MNs of distal muscles (biceps and foot) were in lateral positions (29/33 cells for the biceps and 25/25 cells for the foot muscle) with dendrites extending dorsoventrally, which were located far from the central canal (*Figure 6—figure supplement 1E*). Some of the axial MN dendrites extended to mCherry[+] CSF-cNs and their presumptive ascending axons

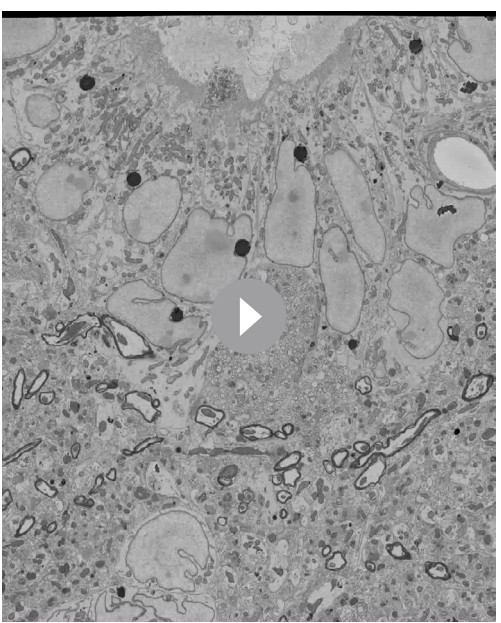

**Video 5.** Serial section serial block-face scanning electron microscopy images of cerebrospinal fluid-contacting neurons and the subependymal axon bundle around the central canal.

https://elifesciences.org/articles/83108/figures#video5

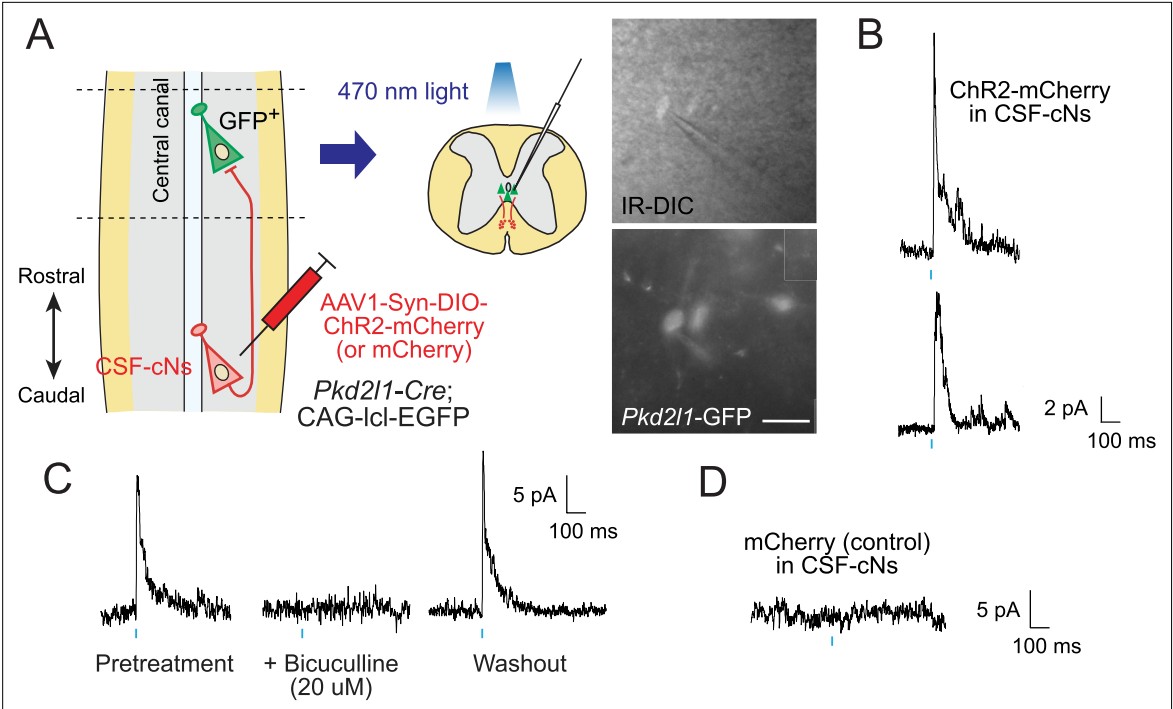

**Figure 5.** Electrophysiological recording with optogenetic stimulation reveals recurrent cerebrospinal fluid-contacting neuron (CSF-cN) connections. (**A**) Diagram of electrophysiological recording of *Pkd2l1*-GFP+ CSF-cNs and blue light stimulation of ChR2-mCherry+ axons of caudally located CSF-cNs in spinal slices of *Pkd2l1-Cre*;CAG-lcl-EGFP mice. ChR2-mCherry was introduced by local injection of adeno-associated virus (AAV)-Syn-DIO-ChR2-mCherry (or AAV-Syn-DIO- mCherry for controls) into the caudal thoracic cord. The right panels are images of patched *Pkd2l1*-GFP+ CSF-cNs. Scale bar, 25 µm. (**B**) Representative inhibitory postsynaptic current (IPSC) responses of two recorded *Pkd2l1*-GFP+ CSF-cNs evoked by a blue light stimulation (10 ms, blue). (**C**) Representative IPSCs of *Pkd2l1*-GFP+ CSF-cN, blocked under 20 µM bicuculline treatment. (**D**) No IPSCs were evoked in the control spinal cord injected with AAV-Syn-DIO-mCherry. Blue bars indicate blue light pulses.

at the midline and were closely contacted with presynaptic GAD65+ or postsynaptic gephyrin+ puncta (*Figure 6M and N*, *Figure 6—figure supplement 1F, G*). We further found that GFP+ dendrites occasionally extended and contacted in close to or within mCherry+ ventral fiber bundles of CSF-cNs, with GAD65+ or gephyrin+ puncta (*Figure 6O*, *Figure 6—figure supplement 1H, I*). These GFP+ MN neurites had scarce presynaptic signals of VAChT and Vglut2 (*Nishimaru et al., 2005*), although a weak VAChT staining was occasionally observed in large terminals near the central canal (data not shown). Furthermore, an axon initial segment marker ankyrin-G was only detected in laterally directed GFP+ fibers (*Figure 6—figure supplement 1J*). These suggest that the MN neurites extending medially to the central canal are dendrites and receive inhibitory CSF-cN inputs.

We further examined anatomical connections with Chx10+ V2a premotor interneurons, which engage in locomotion and skilled motor behaviors (*Azim et al., 2014*; *Crone et al., 2008*; *Crone et al., 2009*; *Ueno et al., 2018*). In *Chx10Cre*;CAG-lcl-EGFP mice, we observed that *Chx10*-EGFP+ neurites also ran along or across the mCherry+ fiber bundles in the ventral funiculus and were occasionally contacted with GAD65+ puncta (*Figure 6P-S*).

We reanalyzed the SBF-SEM data of the ventral bundle, and found that some of the presynaptic CSF-cN terminals containing COX4-dAPEX2+ mitochondria contacted COX4-dAPEX2− neurites emerging from the outside of the bundle (*Figure 6T*, *Figure 6—figure supplement 1K*). TEM analyses further identified synaptic structures in this region (*Figure 6U*). We also frequently found rosette-like structures in which a COX4-dAPEX2− neurite was surrounded by multiple bouton-like presynaptic terminals containing COX4-dAPEX2+ mitochondria along the rostrocaudal axis; this observation was also confirmed in TEM analyses (*Figure 6V*, *Figure 6—figure supplement 1M*). These boutons contained abundant synaptic vesicles, as well as larger vesicles approximately ~100 nm in size. These types of connection were also found in the subependymal bundle (*Figure 6W*, *Figure 6—figure supplement 1L*). COX4-dAPEX2+ pre-synapses also contacted COX4-dAPEX2− neurites in the area

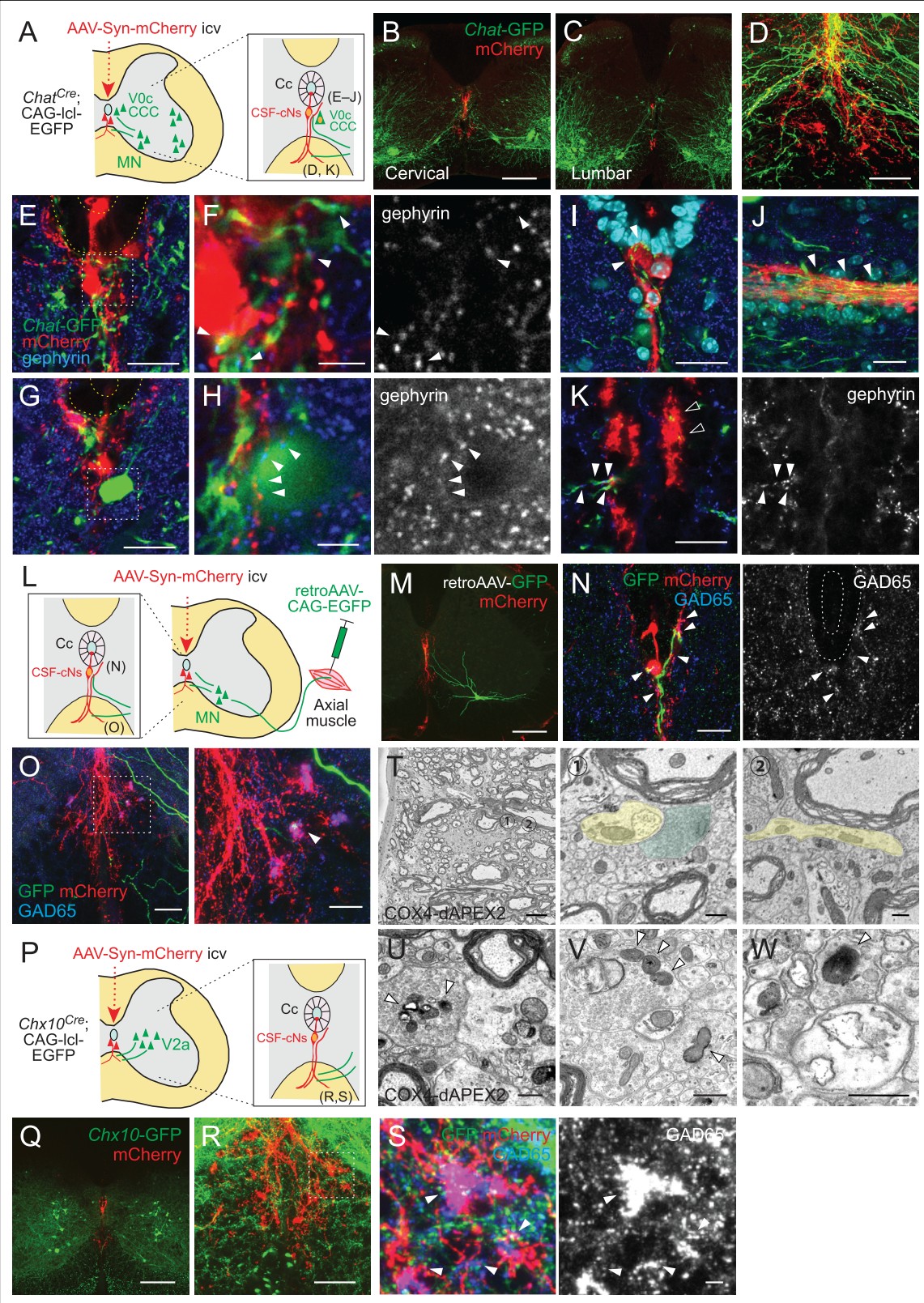

**Figure 6.** Cerebrospinal fluid-contacting neuron (CSF-cN) connections with premotor and motor neurons in the spinal cord. (**A**) Diagram of *Chat*-GFP⁺ cholinergic neurons and adeno-associated virus (AAV)-Syn-mCherry labeled CSF-cNs in *Chat^Cre*;CAG-lcl-EGFP mice. Cc, central canal; CCC, central canal cluster cells; MN, motor neuron. Image positions of (**D–K**) are indicated in the right panel. (**B, C**) Representative images of mCherry⁺ CSF-cNs (red) and *Chat*-GFP⁺ neurons (green) in the cervical (**B**) and lumbar cord (**C**) of *Chat^Cre*;CAG-lcl-EGFP mice. (**D**) A magnified image of *Chat*-GFP⁺

*Figure 6 continued on next page*

*Figure 6 continued*

(green) and mCherry⁺ fibers (red), which are intermingled at the ventral midline and ventral funiculus of the cervical cord. The dotted line indicates the border between the gray and white matter. (**E, F**) *Chat*-GFP⁺ neurites (green) extend near the central canal (yellow dotted lines) (**E**), and those with gephyrin⁺ puncta (blue and white, arrowheads) are contacted by mCherry⁺ CSF-cN fibers (red) (**F**). (**F**) is a magnified view of the white dotted area in (**E**). (**G, H**) *Chat*-GFP⁺ interneuron (green) near the central canal (yellow dotted lines) is contacted by mCherry⁺ CSF-cN fibers (red) apposing gephyrin⁺ puncta (blue and white, arrowheads). (**H**) is a magnified view of the white dotted area in (**G**). (**I, J**) *Chat*-GFP⁺ neurites intermingled with the mCherry⁺ subependymal axon bundle (red) in the cervical cord (arrowheads). Transverse (**I**) and horizontal sections (**J**). (**K**) *Chat*-GFP⁺ neurites (green) are intermingled with the mCherry⁺ axon bundle (red) in the ventral funiculus (arrowheads), with some making contact with the gephyrin⁺ puncta (blue and white, closed arrowheads). (**L**) Diagram representing axial MNs and CSF-cNs, labeled by retroAAV-CAG-EGFP and AAV-Syn-mCherry injections into the axial muscle and lateral ventricle, respectively. Image positions of (**N, O**) are indicated. (**M**) GFP⁺ MN of the dorsal neck muscle (semispinalis capitis) is labeled with retroAAV-CAG-EGFP (green) and mCherry⁺ CSF-cNs at the cervical level. Note the GFP⁺ fibers extending to the central canal, ventral midline, and ventral funiculus. (**N**) GFP⁺ MN neurites extending near the central canal and are contacted by mCherry⁺ CSF-cN fibers with GAD65⁺ puncta (blue and white, arrowheads). (**O**) GFP⁺ MN neurites (green) are intermingled with the mCherry⁺ axon bundle (red) in the ventral funiculus with GAD65⁺ puncta (blue, white, arrowheads). The right panel is a magnified view of the dotted area in the left panel. (**P**) Diagram of *Chx10*-GFP⁺ V2a neurons and AAV-Syn-mCherry-labeled CSF-cNs in *Chx10^Cre^*;CAG-lcl-EGFP mice. Image positions of (**R, S**) are indicated. (**Q**) *Chx10*-GFP⁺ V2a neurons (green) and mCherry⁺ CSF-cNs (red) in the cervical cord of *Chx10^Cre^*;CAG-lcl-EGFP mice. (**R, S**) *Chx10*-GFP⁺ fibers (green) extending to mCherry⁺ fiber bundles (red) in the ventral funiculus. (**S**) shows a magnified view of the dotted area in (**R**), which consists of *Chx10*-GFP⁺ fibers contacted by mCherry⁺ ventral bundles with GAD65⁺ puncta (blue and white) (arrowheads). (**T**) Serial block-face scanning electron microscopy images of the ventral bundles showing COX4-dAPEX2⁺ presynaptic terminals of CSF-cNs (pseudo-colored in green) in contact with a COX4-dAPEX2⁻ neurite coming from outside of the bundle (yellow). Positions 1 and 2 in the left panel are magnified in the right panels (the 80th and 139th serial images). (**U–W**) Transmission electron microscope images of presynaptic terminals containing COX4-dAPEX2⁺ mitochondria (arrowheads) contacting with COX4-dAPEX2⁻ neurites in the bundle of ventral funiculus at the thoracic level (**U, V**) and subependymal area at the cervical level (**W**). Scale bars, 250 μm (**B, C, M, Q**); 50 μm (**D**), (**K**), left panel of (**O**), (**R**); 20 μm (**E**), (**G**), (**I**), (**J**), (**N**), right panel of (**O**); 5 μm (**F**), (**H**), (**S**), left panel of (**T**); 0.5 μm (right panels of **T, U, V, W**).

The online version of this article includes the following figure supplement(s) for figure 6:

**Figure supplement 1.** Connections of cerebrospinal fluid-contacting neurons (CSF-cNs) with premotor and motor neurons (MNs) in the spinal cord.

outside the bundle (*Figure 6—figure supplement 1N*). Although their cell identity could not be determined, the observed COX4-dAPEX2⁻ postsynaptic neurites may correspond to the dendrites of Chat⁺ or Chx10⁺ neurons. Although axo-axonic connections were identified in zebrafish larvae (*Wu et al., 2021*), we could not find axonal morphology in COX4-dAPEX2⁻ neurite, at least in our small serial samples. The recent study suggesting anatomical connections with medial motor column (MMC) MNs and spinal interneurons further supports our observation (*Gerstmann et al., 2022*). Taken together, the data show that CSF-cNs have anatomical connections with motor-related neurons in the spinal cord.

## Inactivation of CSF-cNs induced deficits in treadmill locomotion

Finally, we explored the role of CSF-cNs in motor control. Previous reports in zebrafish larvae have demonstrated the involvement of CSF-cNs in sensorimotor control during swimming (*Böhm et al., 2016*; *Fidelin et al., 2015*; *Hubbard et al., 2016*). To examine their functions in mice, we inhibited the activities of CSF-cNs by introducing hM4Di, designer receptors exclusively activated by designer drugs (DREADD) (*Sternson and Roth, 2014*). We intracerebroventricularly injected AAV-Syn-DIO-hM4Di into *Pkd2l1-Cre* mice, and the mice were then subjected to run on a treadmill before and after treatment with the ligand clozapine-N-oxide (CNO) (*Figure 7A*). The treadmill speed was programmed to increase from 5.0 to 35 m/min (8.33–58.33 cm/s), and it was found that the maximum speed to run was significantly decreased in CNO-treated mice (*Figure 7B*, *Video 6*). We next performed kinematic analyses at a speed of 14 m/min (23.3 cm/s). Overall, the locomotion was not abolished in the CNO-treated mice, in which left-right coordination was not changed (*Figure 7C*). However, we found that step frequencies (Hz) were significantly decreased by CNO treatment, and concomitantly, stride lengths were increased (*Figure 7D–F*, *Video 7*). These changes were not observed in control wild-type (WT) mice treated with AAV and CNO (*Figure 7G–K*). These results suggest that the mice with inactivated CSF-cNs are difficult to keep running on the moving belt, and that CSF-cNs are facultatively involved in locomotor circuitry to control body-limb movement.

We further assessed skilled locomotion in the ladder walk and beam walk tests. After CNO injections, *Pkd2l1-Cre* mice injected with AAV-Syn-DIO-hM4Di showed increased missed steps in the ladder walk test, while they did not show difference in the beam walk test (*Figure 7L–O*). These results suggest that CSF-cNs are involved in sensorimotor control during skilled locomotion.

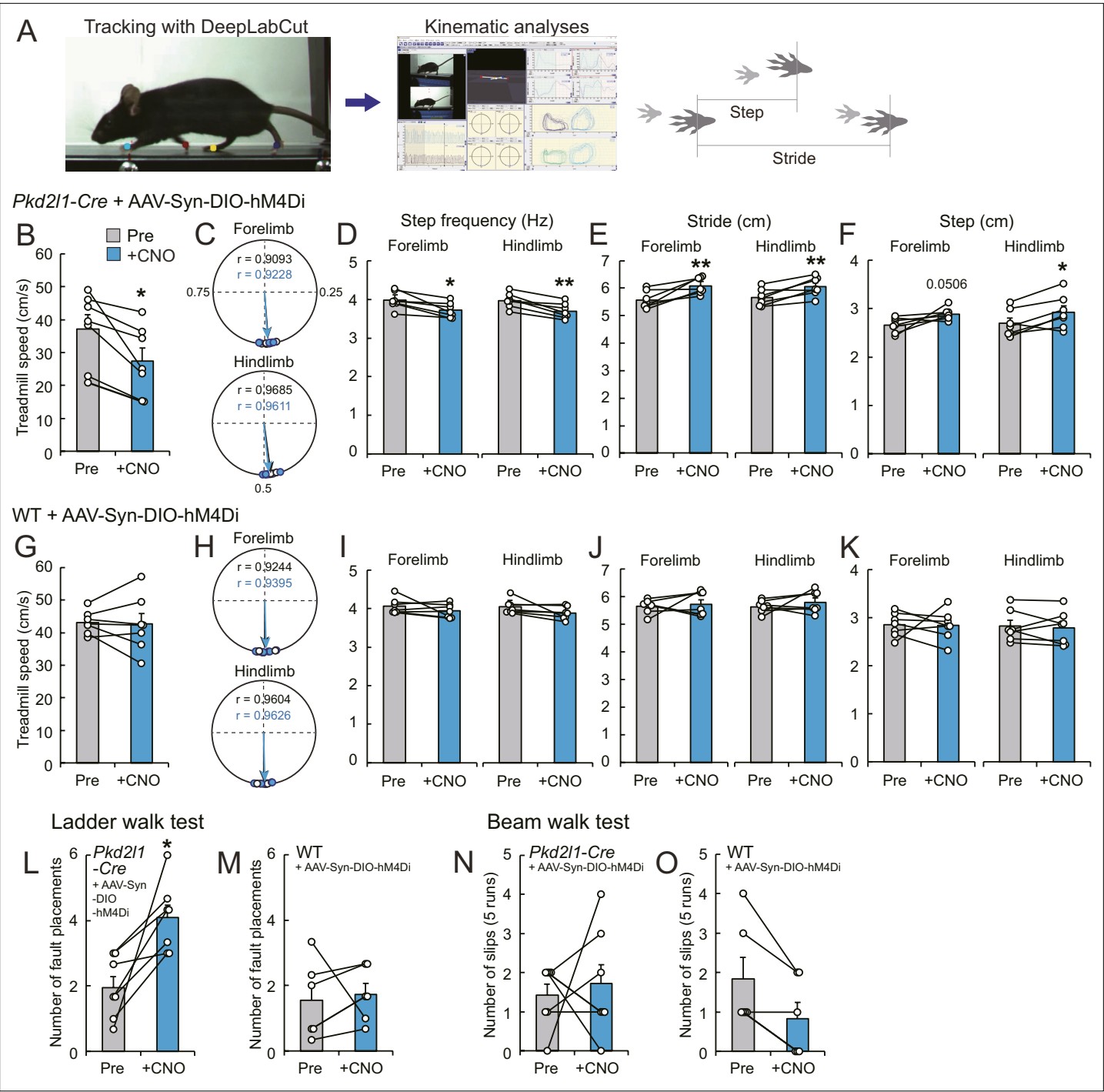

**Figure 7.** Chemogenetic inhibition of cerebrospinal fluid-contacting neurons (CSF-cNs) impairs treadmill locomotion. (**A**) The experimental system to analyze treadmill locomotion. Trajectories of fore- and hind-paw were tracked by DeepLabCut and kinematic parameters including step and stride lengths were analyzed by KinemaTracer. (**B–F**) Analyses of treadmill locomotion pre- and post-inactivation of hM4Di$^+$ CSF-cNs by clozapine-N-oxide (CNO) treatment. *Pkd2l1-Cre* mice were intracerebroventricularly injected with adeno-associated virus (AAV)-Syn-DIO-hM4Di-mCherry to introduce hM4Di-mCherry. (**B**) Maximum treadmill speeds in a program increasing from 8.33 to 58.33 cm/s (5.0–35 m/min). (**C–F**) Circular phase plots representing phase values of the left fore- and hind-paw contacts between two right paw contacts and r (**C**), step frequency (**D**), stride lengths (**E**), and step lengths (**F**) of the left fore- and hindlimbs at a speed of 23.3 cm/s (14 m/min). CNO-treated groups are represented in blue. (**G–K**) Analyses of treadmill locomotion of AAV-Syn-DIO-hM4Di-mCherry-injected wild-type (WT) mice pre- and post-CNO treatment. Maximum treadmill speeds (**G**), circular phase plots (**H**), step frequency (**I**), stride lengths (**J**), and step lengths (**K**) of the left fore- and hindlimbs. (**L, M**) Average number of fault placements in the ladder walk test pre- and post-CNO treatment in AAV-Syn-DIO-hM4Di-mCherry-injected *Pkd2l1-Cre* (**L**) and WT mice (**M**). (**N, O**) The number of slips in

*Figure 7 continued on next page*

*Figure 7 continued*

five runs of the beam walk test pre- and post-CNO treatment in AAV-Syn-DIO-hM4Di-mCherry-injected *Pkd2l1-Cre* (**N**) and WT mice (**O**). Paired t-test (**B**,**D–F**, **G**, **I–M**), Wilcoxon matched-pairs signed rank test (**N, O**); *p<0.05, ** p<0.01; the mean ± standard error of the mean (SEM); n=7 (**B–L, N**), n=6 (**M, O**).

The online version of this article includes the following source data and figure supplement(s) for figure 7:

**Source data 1.** Raw data for maximum treadmill speeds, circular phase plots, step frequency, stride lengths, step lengths, the ladder walk test, and the beam walk test.

**Figure supplement 1.** Schematic model of cerebrospinal fluid-contacting neuron (CSF-cN) connections in the spinal circuits.

## Discussion

In the present study, we developed a novel methodology which enabled genetic labeling and manipulation of CSF-cNs in mice. Although the wide distribution of CSF-cNs in the spinal cord has implied that they have important physiological roles in mammals, their connections and functions have remained poorly understood. Our developed methods proved to be effective to analyze CSF-cN structure and function, and we uncovered typical caudo-rostral projections and functional connections in the spinal circuitry that are involved in locomotor control.

### Specific methods to label and manipulate CSF-cNs

Although intracerebroventricular AAV injections have been widely used, especially for gene delivery (*Dirren et al., 2014*; *Galvan et al., 2021*), no studies have investigated their applicability for CSF-cN labeling. Some previous studies have reported the intracerebroventricular or spinal injection of traditional neural tracers for the purpose of labeling ventricular cells, including CSF-cNs (*Jalalvand et al., 2014*; *Song et al., 2019*; *Song and Zhang, 2018*). Nevertheless, the AAV-mediated method can express any genes of interest primarily into CSF-cNs; this increases specificity, depth, and diversity of the analyses. Since we noticed that AAV8 was not efficient in labeling CSF-cNs (data not shown) and some other serotypes spread throughout the central nervous system (*Galvan et al., 2021*; *Mathiesen et al., 2020*), an appropriate serotype as well as a specific promoter would be essential to successfully target CSF-cNs. Although a few unrelated spinal neurons and fibers were occasionally labeled, possibly due to a leak or transsynaptic transfer of AAV1 (*Zingg et al., 2017*), the combination of *Pkd2l1-Cre* was able to further restrict labeling in PKD2L1$^+$ neurons that contacted the CSF. This approach excluded other ventricular cells, as well as ectopic PKD2L1$^+$ cells located in the ventromedial spinal

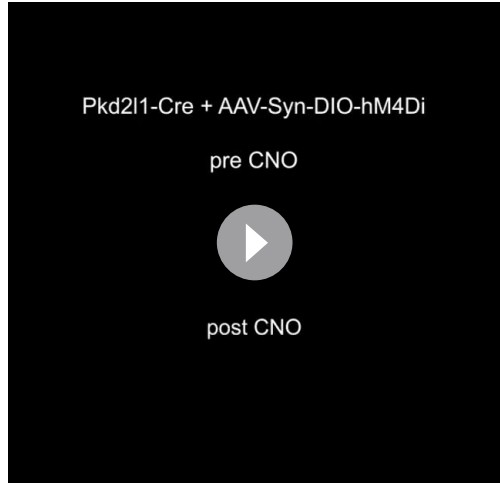

**Video 6.** Treadmill locomotion during a programmed increase in speeds from 5.0 to 35 m/min in an hM4Di-introduced *Pkd2l1-Cre* mouse pre- and post-clozapine-N-oxide (CNO) treatment.

https://elifesciences.org/articles/83108/figures#video6

**Video 7.** Treadmill step frequency at a speed of 14 m/min in an hM4Di-introduced *Pkd2l1-Cre* mouse pre- and post-clozapine-N-oxide (CNO) treatment.

https://elifesciences.org/articles/83108/figures#video7

cord (*Tonelli Gombalová et al., 2020*; *Jurčić et al., 2021*) and *Pkd2l1*[+] oligodendrocytes in the gray matter. A recent study reported that *Pkd2l1*[IRES-Cre] mice recombined undetermined cell types, in which ~84% of *Pkd2l1*-GFP[+] cells corresponded to CSF-cNs (*Gerstmann et al., 2022*). Thus, the present viral tracing method can provide higher specificity, resolution, and flexibility to introduce (or reduce) specific genes or genetic tools, which will further contribute to studies of CSF-cNs.

## Structures of CSF-cNs revealed by viral tracing

Our methodological approach was able to demonstrate, for the first time, the origin and unique structure of mouse CSF-cNs fibers, which projected ventro-rostrally and back to the central canal. Although each CSF-cN projection showed some variance, the following common features were observed: (1) a thin fiber derived from the soma extends ventrally, where bouton structures are scarce; (2) multiple short neurites also extend locally; (3) ventral axons extend rostrally in the ventral funiculus with multiple boutons; (4) ventral axons form several collaterals dorsally, which project back to the central canal; and (5) the branches extended rostrocaudally after reaching the central canal, which also has multiple boutons.

It has been reported that CSF-cNs have ipsilateral ascending tracts in two to six segments in zebrafish larvae (*Djenoune et al., 2017*; *Fidelin et al., 2015*) and in frogs (*Dale et al., 1987*), suggesting conserved projection features among species. The rostral extensions of over 1800–4800 μm observed in our study would correspond to at least over one to four spinal segments in mice. The axons did not show clear commissural innervation until they extended to the ventral funiculus; however, dorsal collaterals often exhibited bilateral innervation around the central canal. The possibly conserved ipsilateral projections should be further examined regarding functional connections on either side, along with those of dorsolateral CSF-cNs, which were not effectively analyzed in the present study. In mammals, the origins of ventral axon bundles were suggested by the expression of P2X2 and PSA-NCAM, particularly in the caudal to lower thoracic cord in rats (*Stoeckel et al., 2003*). Recent analyses of *Pkd2l1-Cre* and reporter mice have also suggested the presence of rostrocaudally or caudorostrally running bundles (*Jurčić et al., 2021*). Our present data revealed the origin and predominant rostral direction of ventral axons at a single-cell and ultrastructural levels. Although P2X2[+] fibers have been reported along the central canal of the thoracic cord in rats (*Stoeckel et al., 2003*), we demonstrated for the first time that the subependymal bundles also originate from CSF-cNs and exist at the cervical level.

Our method also enabled to visualize the apical side, in which dendritic bulbs extended to the CSF and occasionally contained cilia-like protrusions. Previous slice experiments have suggested that the bulb and cilia can detect mechano- or chemosensory information (e.g. pH) in mice (*Huang et al., 2006*; *Orts-Del'Immagine et al., 2016*; *Orts-Del'immagine et al., 2012*). Recent studies have shown that the cilia of CSF-cNs are required to detect spinal bending and maintenance of spinal curvature in zebrafish larvae (*Orts-Del'Immagine et al., 2020*; *Sternberg et al., 2018*) and for proper skilled movements in mice (*Gerstmann et al., 2022*), thus supporting the critical role of these neurons in the mechanosensory system. Electron microscopic analyses have further revealed that close vicinity of the bulb to the Reissner fiber enables the detection of spinal curvature and maintenance of a straight body axis in zebrafish larvae (*Cantaut-Belarif et al., 2018*; *Orts-Del'Immagine et al., 2020*; *Troutwine et al., 2020*). Although the Reissner fiber was shown to be present in rodents (*Aboitiz and Montiel, 2021*; *Bjugn et al., 1988*), however, its role has not yet been determined. The present methodological approach will facilitate the exploration of CSF-cN sensory functions in mice in future studies.

## Connectivity of CSF-cNs in the spinal circuitry

Elucidation of the connectivity of CSF-cNs would be essential to understand their functions. In this study, we found that CSF-cNs send collaterals around the central canal and are connected to other rostral CSF-cNs. While previous electrophysiological studies reported that CSF-cNs received GABAergic synaptic inputs (*Jurcic et al., 2019*; *Marichal et al., 2009*; *Orts-Del'Immagine et al., 2016*; *Orts-Del'immagine et al., 2012*), the presynaptic partners remained undetermined. We showed direct evidences of rostrocaudal recurrent circuitry by using a single-cell tracing, serial electron microscopic analyses, and electrophysiological recording. These indicate that caudal CSF-cNs are the primary sources of GABAergic inputs. This is supported by previous electron microscopic

observations of P2X2 ⁺ presynaptic terminals, which were present around the central canal (*Stoeckel et al., 2003*). Recent rabies virus-based monosynaptic retrograde tracing further supports our observation (*Gerstmann et al., 2022*). Mutual connections have been also suggested to be present in carps (*Vigh et al., 1974*), implicating a conserved connectivity among species. Other undetermined sources of glutamatergic, cholinergic, and monoaminergic inputs were also reported (*Corns et al., 2015*; *Jurcic et al., 2019*; *Orts-Del'Immagine et al., 2014*; *Orts-Del'Immagine et al., 2016*), and these may also control CSF-cN functions.

Connectivity of CSF-cNs has recently been determined in zebrafish larvae. CSF-cNs are connected with CaP primary MNs that innervate axial musculature and are required for fast locomotion and postural control (*Hubbard et al., 2016*). This is likely partially conserved in mice, based on our anatomical data showing the connections with axial muscle MNs. Zebrafish CSF-cNs are additionally connected to V0-v premotor interneurons, which are involved in slow locomotion (*Fidelin et al., 2015*), and CoPA interneurons involved in sensory-motor gating (*Hubbard et al., 2016*). Although we did not assess relevant subtypes, Chat⁺ and V2a interneurons were found to be anatomically connected with CSF-cN axons. These neurons are known to be recruited in locomotion (*Crone et al., 2008*; *Crone et al., 2009*; *Miles et al., 2007*; *Tillakaratne et al., 2014*; *Zagoraiou et al., 2009*). Recent electrophysiological analyses have shown a functional connection with presumptive V0c neurons, and anatomical analyses further demonstrated presynaptic contacts onto the soma of Lhx1⁺ (V0/dI4) and Chx10⁺ V2a interneurons as well as MMC neurons (*Gerstmann et al., 2022*). Although they showed some different subcellular patterns of contacts from our observations in which the contacts were mainly seen on cell bodies rather than neurites, this discrepancy may be attributed to the use of different labeling methods. Our methods provide high specificity and resolution to additionally determine the anatomical patterns of connectivity with spinal interneurons, which especially showed a unique distribution of contacts with distal dendrites around the central canal as well as within the unmyelinated axon bundles, forming multiple synaptic clusters in a rosette-like structure.

Our anatomical data suggest that CSF-cNs may be involved in a motor module that controls axial muscles (*Figure 7—figure supplement 1*). V2 neurons have presynaptic inputs onto dendrites of axial muscles (*Goetz et al., 2015*), and the medial population of V2a neurons is presumed to be premotor neurons for axial muscle MNs (*Hayashi et al., 2018*). As for cholinergic neurons, lamina X contains several types of cholinergic neurons (*Barber et al., 1984*; *Miles et al., 2007*; *Zagoraiou et al., 2009*). V0c neurons connect to MNs including axial muscles with C boutons and modulate locomotor activities (*Barber et al., 1984*; *Goetz et al., 2015*; *Miles et al., 2007*; *Zagoraiou et al., 2009*). Chat⁺ central canal cluster cells also respond to locomotion (*Barber et al., 1984*; *Tillakaratne et al., 2014*). These neurons may form a motor module in the ventromedial gray matter and be controlled by inhibitory inputs from CSF-cNs. Since electrophysiological recording could not be applied in the present study, functional connections between CSF-cNs and axial MNs, Chat⁺ and V2a interneurons are still hypothetical. Further studies are required to determine the patterns of connections.

We also found strong expressions of inhibitory synaptic proteins in the axon bundles, consistent with synaptic marker expressions in ventral fibers (*Stoeckel et al., 2003*). Our serial electron microscopic analyses additionally revealed that synaptic structures are actually present in the bundle. In particular, the unmyelinated axons often exhibited a typical multiple bouton structures that surrounded and contacted other running dendrites. Although the functions of unmyelinated fiber bundles and their internal synaptic structures are not yet understood, strong inhibitory transmission in this region may have an essential role in spinal function. Axo-axonic connections, which could not be clearly identified in the present study, should also be explored in future studies (*Wu et al., 2021*). Since this region occasionally contains larger vesicles (~100 nm) that are positioned far from the synaptic cleft, volume transmission may also occur here. In addition, structural differences were found among the bundles at cervical to sacral levels. For example, caudal CSF-cNs had more rostrally branched and longer axons in the ventral bundle that were closely aligned to the median fissure, whereas rostral CSF-cNs comprised relatively shorter rostral projections in multiple distributed bundles and more dorsally branched collaterals. These features may have functional implications that should be explored more. It is also unclear whether CSF-cN outputs are biased on specific spinal levels, such as those of central pattern generators (CPGs) for locomotion located at C5–T1 and T12–L5 (*Gordon et al., 2008*; *Kjaerulff and Kiehn, 1996*), or evenly distributed throughout the spinal cord. Dorsal collateral density was higher at the cervical level but lower in the lumbar cord, suggesting that dorsal innervations

might not correlate with the CPG locations. Additionally, CSF-cNs had other connections within the ventral bundles possibly throughout the spinal cord, suggesting their influence on the circuitry such as for axial muscles at various spinal levels. Notably, ablation of rostral and caudal CSF-cNs in zebrafish larvae exhibited different deficits in movements (*Wu et al., 2021*). Further research is required to understand the difference or similarity between functions at different spinal levels.

## Sensorimotor functions of CSF-cNs in locomotion

Our results indicate that CSF-cNs are required for smooth locomotion on the treadmill. Notably, CSF-cNs inactivation decreased maximum speed and step frequency, accompanied by an increase of stride lengths. This effect may be related to observations reported in zebrafish larvae, in which silencing of CSF-cNs induced a decrease in tail-beating frequency, as well as postural defects (*Böhm et al., 2016*; *Hubbard et al., 2016*; *Wu et al., 2021*). Some previous studies predicted a lower contribution of mouse CSF-cNs to general movements, since a decrease in the number of CSF-cNs in adult or in C57Bl/6 N mice did not apparently change motor behaviors (*Tonelli Gombalová et al., 2020*; *Orts-Del'Immagine et al., 2017*). A recent study that deleted *Pkd2l1*-Cre[+] cells with diphtheria toxin, however, reported deficits in skilled movements on a narrow beam and horizontal ladder with wide rungs that were partially replicated in our experiments, although no impairments were observed in walking (*Gerstmann et al., 2022*). Our detailed kinematic analyses combined with specific targeting of CSF-cNs additionally revealed that acute inactivation of CSF-cNs disturbed smooth locomotion on the treadmill to maintain step frequency at a high speed. Nevertheless, the functional effect seemed to be relatively weaker compared to the phenotypes reported in zebrafish larvae (*Böhm et al., 2016*; *Fidelin et al., 2015*; *Hubbard et al., 2016*; *Wu et al., 2021*), as we could not see clear defects in rostrocaudal coordination or postural control. It is possible that alternate circuits for limb control or other mechanosensory systems (e.g. muscle spindles and Golgi tendons that are especially developed in mammals) predominantly work or compensate to maintain locomotion in quadrupedal species. Thus, although it is phylogenetically conceivable that CSF-cNs are involved in a motor module for posture control, it remains unclear if they are fully functional or rudimentary in mice.

A model illustrating CSF-cN circuits for locomotion is presented in *Figure 7—figure supplement 1*. CSF-cNs may interoceptively detect spinal bending to control axial movements. CSF-cNs may subsequently convey inhibitory signals onto a rostral motor module comprising axial MN, Chat[+], and V2a premotor neurons and also inhibit rostral CSF-cNs that sequentially release inhibitions onto a more rostral motor module. Longitudinal transmissions of inhibitory signals are shown to be essential to evoke appropriate locomotor coordination or frequency in several contexts. For example, V1 inhibitory neurons that contain both ascending and descending projections are important to control locomotion speed (*Gosgnach et al., 2006*). V2b inhibitory neurons that predominantly comprise descending projections are required for rostrocaudal flexor-extensor coordination, which is cooperated with V1 inhibitory neurons (*Britz et al., 2015*; *Zhang et al., 2014*). CSF-cNs may share some properties with V2b inhibitory neurons, since they both originate from the p2 domain, as well as p3 and pMN for other types of CSF-cNs (*Petracca et al., 2016*). The inhibitory signals may further influence the activities of excitatory V2a interneurons, which have a role in rhythm generation, primarily at faster locomotor speeds in mice (*Crone et al., 2009*). Zebrafish also have similar ascending and descending inhibitory V1 and V2b circuitry that inhibit MNs and V2a neurons to control swimming speed (*Callahan et al., 2019*; *Kimura and Higashijima, 2019*; *Sengupta et al., 2021*). These rostrocaudal inhibitory transmissions may control sequential axial MNs and muscle activities, which are coupled to limb movements and are essential for mobilization and stabilization during locomotion (*Hinckley et al., 2015*; *Schilling and Carrier, 2010*). Further electrophysiological and electromyographic analyses are required to test the involvement of CSF-cNs in the motor circuitry of axial muscles in mice.

Although CSF-cNs respond to spinal bending in zebrafish, the present study did not examine what CSF-cNs specifically detect to control locomotion. Although mechanosensory signals would be generated during treadmill locomotion or ladder walking, these should be further investigated in future studies. Other chemosensory signals, such as pH changes or compounds proposed to be generated in respiratory changes, excess neural activities, fatigue, autonomic, or immune responses, may also be involved (*Huang et al., 2006*; *Jalalvand et al., 2016b*; *Prendergast et al., 2019*). Such signals are currently poorly understood in mammals.

## Conclusion and future perspectives

Our study may be limited to understand the characteristics of CSF-cN subpopulation in part, because the majority of AAV-labeled cells were located on the ventral side. At least two CSF-cN subpopulations, referred to as CSF-cN' (dorsolateral) and CSF-cN" (ventral), have been reported in zebrafish and mice (*Di Bella et al., 2019*; *Park et al., 2004*; *Petracca et al., 2016*). These two subpopulations exhibit a different connectivity and functions in zebrafish. Ventral CSF-cNs respond to longitudinal contractions, connect to CaP MNs, and are involved in fast swimming, while dorsal CSF-cNs respond to ipsilateral lateral bending, connect to putative V0-v, and are involved in slow swimming circuit (*Böhm et al., 2016*; *Djenoune et al., 2017*; *Hubbard et al., 2016*). Studies in lampreys have also shown that lateral and ventral CSF-cNs have different mechano- and chemosensory properties, as well as the presence of motile cilia (*Jalalvand et al., 2022*). Although their functional and connectional differences remain unknown in mice, our data may be comparable to those of the recent study that labeled whole CSF-cN population by using *Pkd2l1$^{IRES-Cre}$* mice, in which they showed some different connections and behavioral changes from our present data (*Gerstmann et al., 2022*). In this context, the features of the minor bidirectionally or caudally projecting CSF-cNs also remain unknown.

In conclusion, we found a specific method to label and manipulate mouse CSF-cNs and revealed their structures, connectivity, and involvement in locomotion function, which have long been remained poorly understood in mammals. The method developed here will pave the way for a more detailed exploration of the features of mouse CSF-cNs, ranging from connectivity to physiological and molecular functions and will provide further insight into the architecture of spinal circuits that control sensorimotor or other homeostatic functions in mammals.

# Materials and methods

## Animals

The following mice were used in this study: male C57BL/6 J mice (Jackson laboratory), *Pkd2l1-Cre* (a bacterial artificial chromosome transgenic mouse line; a gift from C. Zuker, Columbia University) (*Huang et al., 2006*), *Chat$^{Cre}$* (#006410, Jackson laboratory), *Chx10$^{Cre}$* (a gift from S. Crone, CCHMC, and K. Sharma, University of Chicago) (*Azim et al., 2014*), CAG-lox-CAT-lox-EGFP (a gift from J. Robbins, CCHMC) (*Nakamura et al., 2006*), and *Rosa26$^{CAG-lox-stop-lox-tdTomato}$* mice (Ai14; #007914, Jackson laboratory). All the procedures were performed in accordance with protocols approved by the Institutional Animal Care and Use Committee of Niigata University (SA00656) and ARRIVE guidelines.

## AAV production

AAV plasmids were generated as follows: for pAAV-Syn-mCherry, mCherry of pmCherry-C1 plasmid (Clontech) was subcloned into pAAV-hSyn-EGFP (Addgene, #50465) in a substitution of EGFP; for pAAV-Syn-DIO-COX4-dAPEX2 and pAAV-Syn-DIO-SYP-HRP, COX4-dAPEX2 and SYP-HRP derived from pAAV-EF1a-COX4-dAPEX2 (Addgene, #117176) and pAAV-EF1a-SYP-HRP (Addgene, #117185) were subcloned into pAAV-Syn-DIO-MCS that had multicloning sites (MCS) between the DIO site of pAAV-Syn-DIO-mCherry (Addgene, #50459) in a substitution of mCherry; and for pAAV-Syn-DIO-ChR2-mCherry, ChR2(H134R)-mCherry derived from pAAV-CAG-ChR2(H134R)-mCherry (Addgene, #100054) was subcloned into pAAV-Syn-DIO-MCS plasmid. pAAV-hSyn-EGFP (Addgene, #50465), pAAV-CAG-EGFP (pENN.AAV.CB7.CI.eGFP.WPRE.rBG; Penn vector core, #AV-1-PV1963), and pAAV-Syn-DIO-mCherry (Addgene, #50459) were further used for AAV production.

Human embryonic kidney 293T cells (632273, TAKARA) were used to produce AAVs. The cell identity was confirmed by short tandem repeat profiling, and the cells were tested negative for mycoplasma contamination. They were transfected with pAAV plasmid, pAAV2/1 (Penn vector core) or rAAV2-retro helper (Addgene, #81070 *Tervo et al., 2016*), and pHelper plasmid (TAKARA). The supernatants were purified and concentrated in phosphate-buffered saline (PBS) containing 0.001% Pluronic F68. The titer of AAV was determined with a real-time PCR thermal cycler (Dice, TAKARA). The resultant AAV1-Syn-mCherry ($2.9 \times 10^{12}$ GC/ml), AAV1-Syn-EGFP ($5.0 \times 10^{12}$ GC/ml), AAV1-Syn-DIO-COX4-dAPEX2 ($6.3 \times 10^{12}$ GC/ml), AAV1-Syn-DIO-SYP-HRP ($3.6 \times 10^{12}$ GC/ml), retroAAV-CAG-EGFP ($9.7 \times 10^{12}$ GC/ml), AAV1-Syn-DIO-ChR2(H134R)-mCherry ($3.4 \times 10^{12}$ GC/ml), and AAV1-Syn-DIO-mCherry ($4.1 \times 10^{12}$ GC/ml) were stored at $-80°C$ until use. AAV1-CAG-tdTomato ($2.8 \times 10^{12}$ GC/ml; Penn vector core, #AV-1-PV2126), AAV1-Syn-GFP-Cre ($0.43–3.2 \times 10^{11}$ GC/ml; Penn vector core, #AV-1-PV1848),

and AAV1-Syn-DIO-hM4Di-mCherry (2.59 × 10^{12} GC/ml; Addgene, #44362-AAV1) were obtained from Penn vector core or Addgene.

## AAV injection and anterograde and retrograde tracing

Injections were performed as previously reported (*Ueno et al., 2018*). Mice at 6 wk for histological analyses and at 4 wk for behavioral analyses were anesthetized and placed in a stereotaxic frame. The scalp was incised, and small holes were made on the skull at the corresponding injection sites by using a 27 G needle. AAV was injected into the left lateral ventricle (depth of 1.5 mm; coordinates, 0 mm posterior, 1.0 mm lateral to the bregma; total volume, 0.6 μl) using a Hamilton syringe tipped with a glass micropipette. The mice were perfused 2 wk later for histological analyses.

For retrograde tracing of MNs, retroAAV-CAG-EGFP (9.7 × 10^{12} GC/ml) was injected into the following muscles on the right side: semispinalis capitis at the dorsal neck; sternocleidomastoid at the ventral neck; transversospinales at the caudal thoracic to lumbar level (*Brink and Pfaff, 1980*) (three sites; total volume, 1 μl); biceps of the forelimb (total 2.0 μl), and dorsal muscles of the foot including the extensor digitorum brevis (total 0.5 μl) at P14.

## Immunohistochemistry

The animals were perfused transcardially with 4% paraformaldehyde (PFA) in 0.1 M phosphate buffer (PB). The brain and spinal cord were dissected and postfixed in the same fixatives overnight. The tissues were then cryopreserved in 30% sucrose in PBS overnight and embedded in Tissue-Tek OCT compound (Sakura Finetek). Serial 20- or 50-μm-thick sections were made with a cryostat and mounted on MAS-coated slides (Matsunami). For immunohistochemistry, the sections were blocked with 1% bovine serum albumin (BSA) in 0.3% Triton X-100/PBS for 2 h and then incubated with primary antibodies overnight at 4°C. After washing with 0.1% Tween-20/PBS, the sections were incubated with corresponding secondary antibodies for 2 h at room temperature (RT).

For the analyses of anatomical connections of spinal neurons and CSF-cNs, 80-μm-thick floating sections were blocked with 1% BSA/0.3% Triton X-100 in PBS for 1 h, followed by incubation with primary antibodies in 0.1% Triton X-100/PBS for three overnights at 4°C. After washing with 0.1% Triton X-100 in PBS, sections were incubated with corresponding secondary antibodies for three overnights at 4°C.

The following primary antibodies were used: goat anti-mCherry (1:500 AB0040, Sicgen), rabbit anti-RFP antibody (1:1000; 600-401-379, Rockland), rabbit anti-PKD2L1 (1:500; kindly gifted from Dr. Matsunami, Duke University) (*Ishimaru et al., 2006*), rabbit and guinea pig anti-PKD2L1 (1:1000 and 1:500, respectively; generated by immunization with PKD2L1 peptide 731–749: C-KLKMLERK-GELAPSPGMGE conjugated with KLF, in accordance with the previous study) (*Ishimaru et al., 2006*), guinea pig anti-GAD65 (1:1000; 198–104, SYSY), mouse anti-GAD67 (1:1000; MAB5406, Millipore), guinea pig anti-VGAT (1:500; 131–004, SYSY), guinea pig anti-Vglut2 (1:10000; Millipore, AB2251), goat anti-VAChT (1:1000; Millipore, ABN100), mouse anti-gephyrin (1:1000; 147–021, SYSY), rabbit anti-GFP (1:1000, A11122, Invitrogen), rat anti-GFP (1:1000, 04404–84, Nacalai), rabbit anti-Olig2 (1:500; AB9610, Millipore), rabbit anti-Sox9 (1:500; 82630 S, Cell signaling), and rabbit ankyrin-G (1:500; Frontier institute, Af610). For secondary antibodies, Alexa Fluor 488, 568, or 647 donkey anti-rabbit, mouse, rat, goat, or guinea pig IgG antibody (1:1000; Invitrogen or Jackson ImmunoResearch) were used.

The sections were then washed with PBS and counterstained with 4',6-diamidino-2-phenylindole (DAPI; 1 μg/ml; Santa Cruz Biotechnology) to visualize nuclei. The images were acquired with a fluorescence microscope (Olympus, BX51) or a confocal microscope (Olympus, FV3000). For histological evaluation, PKD2L1+mCherry+/mCherry+ cells were counted in 24 spinal cord sections of 20-μm-thickness at C3–5, T4–7, and L1–3 levels. To assess the anatomical connections of MNs labeled with retro AAVs, only GFP+ neurites traced from the soma of the appropriate MMC were used. For ChR2-mCherry-labeling of CSF-cN fibers in electrophysiological experiments, the number of mCherry+ axons at the ventral funiculus of the recorded slices immunostained with an anti-mCherry antibody was counted in images obtained with a confocal microscope.

## Tissue clearing and 3D imaging for single-cell tracing

Tissue clearing was performed by CUBIC with minor modifications (*Tainaka et al., 2018*). Postfixed spinal cords of WT mice injected with AAV-Syn-mCherry (2.9 × 10^{12} GC/ml; for whole tracing) or

*Rosa26^{lsl-tdTomato}* mice injected with AAV-Syn-Cre (0.43–3.2 × 10$^{11}$ GC/ml; for single-cell tracing) and AAV-Syn-EGFP (5.0 × 10$^{12}$ GC/ml; for whole tracing) were dissected sagittally at the midline area including the central canal at a thickness of approximately 1 mm and washed three times in PBS. For delipidation and decoloring, the tissues were incubated in CUBIC-L (10 wt% *N*-butyldiethanolamine, 10 wt% Triton X-100) for two overnights at RT. After three washing cycles for 2 h each with PBS, the tissues were incubated in the blocking buffer (5% donkey serum, 0.5% Triton X-100, in 0.05% NaN$_3$ / PBS) for 12 h, followed by incubation with primary antibodies in blocking buffer at 4°C for four overnights. Rabbit anti-RFP (1:1000; 600-401-379, Rockland) and rat anti-GFP (1:1000, 04404–84, Nacalai) were used for primary antibodies. They were then washed in 0.05% NaN$_3$/PBS overnight, and subsequently, incubated with secondary antibodies (Alexa Fluor 568 anti-rabbit and Alexa Fluor 488 anti-rat IgG antibodies), and DAPI (1 μg/ml) in blocking buffer at 4°C for four overnights. They were then washed three times in 0.05% NaN$_3$/PBS for 1 h each and then post-fixed in 1% PFA at RT for 12 h. For refractive index (RI) matching, they were incubated in 50% CUBIC-R (45 wt% antipyrine, 30 wt% nicotinamide, and pH 8–9 adjusted by *N*-butyldiethanolamine) for 6 h and then in CUBIC-R for 24 h at RT. The cleared tissues were imaged under a confocal microscopy (Olympus FV-3000). The obtained images were reconstructed in IMARIS software, and the labeled cells and axons were traced by using the Filament Tracer tool. A total of 25, 28, 10, and 8 cells which projections were mostly traced in the cervical, thoracic, lumbar, and sacral levels, respectively (total 17 mice), were assessed for rostrocaudal axon lengths, and the number of collaterals projecting to the dorsal direction and rostrocaudal direction in the ventral funiculus. The projection side was examined in three structural parts; (1) ventral projections to the ventral funiculus, (2) rostral projections in the ventral funiculus, and (3) dorsal projections to the central canal. Other 62, 81, 85, and 33 cells from the cervical, thoracic, lumbar, and sacral levels, respectively, were traced at least until the axons reached the ventral funiculus; therefore only the rostrocaudal directions of projections were evaluated in these cases.

After observation, some of the tissues were washed with PBS and embedded in 5% low-melting agarose/PBS. They were sectioned at 80 μm in a transverse plane by using a vibratome (Leica). The floating sections were stained with rabbit anti-RFP (1:1000; 600-401-379, Rockland), rat anti-GFP (1:1000, 04404–84, Nacalai), and guinea pig anti-GAD65 (1:1000; 198–104, SYSY) for primary antibodies; Alexa Fluor 568 anti-rabbit, Alexa Fluor 488 anti-rat (1:1000; Invitrogen), and Alexa Fluor 647 anti-guinea pig IgG (1:1000; Jackson ImmunoResearch) for secondary antibodies. The sections were imaged under confocal microscopy (Olympus FV3000).

## SBF-SEM and TEM analyses with peroxidase-based organelle labeling

Spinal cord slices of *Pkd2l1-Cre* mice injected with AAV-Syn-DIO-SYP-HRP or AAV-Syn-DIO-COX4-dAPEX2 at 6 wk were stained with DAB reactions and then subjected to SBF-SEM analyses, as previously described with minor modifications (*Nguyen et al., 2016*; *Thai et al., 2016*; *Zhang et al., 2019*). Four weeks after AAV injection, mice were perfused with 2.5% glutaraldehyde and 2% PFA in 0.1 M PB (pH 7.4), and spinal cords were dissected and postfixed in the same fixative for 2 h at 4°C. Transverse sections were then made by a vibratome (100 μm thick; Leica). The sections were incubated in 0.03% DAB for 2 h and then in a 0.003% H$_2$O$_2$ and 0.03% DAB solution for 2 h. The sections were then post-fixed in 2.5% glutaraldehyde and 2% PFA in 0.1 M PB (pH 7.4) at 4°C for one overnight.

The sections were dissected using an ophthalmic knife under a stereomicroscope. The blocks were then treated with 2% OsO$_4$ and 1.5% potassium ferricyanide in PBS for 1 h at 4°C, 1% thiocarbohydrazide for 20 min at RT, 2% aqueous OsO$_4$ for 30 min at RT, 1% uranyl acetate solution overnight at 4°C, and a lead aspartate solution for 2 h at 50°C. The samples were subsequently dehydrated in a graded ethanol series (60, 80, 90, and 95%) and infiltrated with dehydrated acetone. The sections (1 mm × 1 mm ×200 μm) were embedded in conductive resin containing Durcupan and 5% Ketjen black powder (*Nguyen et al., 2016*) and incubated at 70°C for five overnights to ensure polymerization. The samples were mounted on aluminum rivets, trimmed and observed with SBF-SEM or TEM.

SBF-SEM image acquisition of spinal sections was performed with a Merlin and Sigma scanning electron microscope (Carl Zeiss) equipped with a 3View in-chamber ultramicrotome system (Gatan). Serial image sequences were approximately 20–60 × 40–60 μm wide (5.0 nm/pixel) and >25 μm deep at 50 nm steps. Sequential images were processed using Fiji software (https://imagej.net/software/fiji/). Segmentation of the cell membrane was performed with Microscopy Image Browser software. For 3D reconstruction, Amira software (Thermo Fisher Scientific) was used.

For TEM analyses, ultrathin sections were produced from the samples for SBF-SEM observation with an ultramicrotome (Ultracut UCT, LEICA). These sections were mounted on copper grids and observed with HT7700 (Hitachi High-Tech Co.).

## Whole-cell patch clamp recordings with optogenetic stimulation

AAV-Syn-DIO-ChR2(H134R)-mCherry ($3.4 \times 10^{12}$ GC/ml, 0.2 µl) or AAV-Syn-DIO-mCherry ($4.1 \times 10^{12}$ GC/ml, 0.2 µl) was injected into T9 level of *Pkd2l1-Cre*;CAG-lcl-EGFP mice at P9. Each mouse (P28–31) was deeply anesthetized and perfused intracardially with an ice cold and oxygenated (95% $O_2$/5% $CO_2$) slice solution (250 mM sucrose, 2.5 mM KCl, 25 mM $NaHCO_3$, 1 mM $NaH_2PO_4$, 6 mM $MgCl_2$, 0.5 mM $CaCl_2$, and 25 mM glucose). Following laminectomy and spinal cord extraction (T4–8), thoracic cord coronal slices (300 µm) were cut using a microslicer (Linear Slicer PRO 7; Dosaka). The slices were recovered in oxygenated slice solution at 35°C for 15 min and then maintained in an incubation chamber with oxygenated artificial CSF (ACSF) (117 mM NaCl, 3.6 mM KCl, 2.5 mM $CaCl_2$, 1.2 mM $MgCl_2$, 1.2 mM $NaH_2PO_4$, 25 mM $NaHCO_3$, and 11.5 mM glucose; pH 7.4±0.5, adjusted with HCl) at RT for 30 min before recording.

For recording, slices were transferred to a chamber (ACSF 4 mL min⁻¹, RT). Electrodes were pulled from borosilicate glass capillary tubes (1.5 mm outer diameter borosilicate glass; World Precision Instruments) using a horizontal pipette puller (P-97, Sutter Instruments). The electrodes were filled with a cesium-based internal solution (110 mM Cs-sulfate, 0.5 mM $CaCl_2$, 2 mM $MgCl_2$, 5 mM EGTA, 5 mM HEPES, and 5 mM ATP-Mg salt; pH 7.2, 295–305 mOsm) for voltage-clamp recording. The resistance of the patch pipette was 4–6 MΩ.

The slices were visualized with a fixed upright microscope (E600FN; Nikon) equipped with a water immersion lens (40×/0.8 W) and an infrared-sensitive CCD camera (C2400-79H; Hamamatsu Photonics, Hamamatsu). Signals were amplified with an Axopatch 200B amplifier (Molecular Devices), filtered at 2 kHz, and digitized at 5 kHz. Data were stored and analyzed using the pCLAMP 10.3 data acquisition program (Molecular Devices). After the whole-cell configuration was established, neurons were held in a voltage-clamp at 0 mV for recording IPSCs. To elicit the evoked IPSCs, photostimulation with a 470-nm blue light was used with an LED driver (LEDD1B - T-Cube LED driver, Mounted LED for microscope, Thorlabs), which was passed through a filter (LED-DA/FI/TX-A-000, Optline) and an immersion objective (40×/0.8 W, Nikon) directly above the spinal cord slice (distance, less than 3 mm). The stimuli (duration, 10 ms; frequency, 0.3 Hz; repetition, 10 times; light intensity, ~5 mW/mm²) were controlled with an STO mk II stimulator (BRC). $GABA_A$ receptor-evoked IPSCs were isolated by bath perfusion of 20 µM Bicuculline (Sigma).

## Treadmill locomotor test and DREADD experiments

*Pkd2l1-Cre* or WT mice were randomly selected and injected with AAV-Syn-DIO-hM4Di-mCherry intracerebroventricularly at P28 and were then trained to run on the treadmill (Melquest) twice at week 3 after AAV injection. After 4 wk, mice were subjected to run on the treadmill at the speeds programmed to increase from 5.0 to 35 m/min (8.33–58.33 cm/s) in 1 min. The maximum speed at which the mice could keep running on the treadmill was recorded blindly (n=7 for *Pkd2l1-Cre*; n=7 for WT; *Ueno and Yamashita, 2011*). The following criteria were used for the endpoints of running: (1) the hip touched on the back curtain for more than 3 s; (2) a caudal half of the body was kept under the curtain for more than 3 s; or (3) more than half of the body was under the curtain. The tests were performed three times pre- and 30 min post-injection of CNO (5.0 mg/kg body weight, i.p.; Sigma), and the average speed was calculated. Runs at 14 m/min (23.3 cm/s) were further recorded with high-speed video cameras (198 f/s) set on the left and back of the treadmill and MotionRecorder software of the Kinematracer system (Kissei comtec; *Ueno et al., 2018*; *Ueno and Yamashita, 2011*; *Figure 7A*) (n=7 for *Pkd2l1-Cre*; n=7 for WT). The movies were then transported to DeepLabCut, a markerless pose estimation system based on transfer learning with a deep neural network (*Mathis et al., 2018*), to track the movement of the fore- and hind-paws. The positions of fore- and hind-paws in 280 training frames of the representative mouse locomotion movie were manually labeled, and then the deep neural network was trained with the datasets for 1,030,000 iterations with an NVIDIA GeForce RTX 2070 SUPER Graphics Card. The fore- and hind-paws in the movies were then tracked, and the obtained x-y coordinates were transported to 3DCalculator software of Kinematracer system, which was modified to integrate CSV files generated in DeepLabCut. Stride and step lengths of the left

fore- and hindlimbs (total 28–65 steps in two 11 s movies per mouse), step frequency (Hz), and phase values (ϕ) were then assessed in the KineAnalyzer software. Step frequency (Hz) was determined by the duration of step cycle, which corresponds to the phase between two successive foot contacts of the limb (*Leblond et al., 2003*; *Lemieux et al., 2016*; *Ueno and Yamashita, 2011*). A phase value (ϕ) corresponded to the time point of left fore- or hind-paw contacts relative to the right fore- or hindlimb step cycle. A phase value of 0 or 1 indicates an in-phase coupling (synchrony), while 0.5 reflects an anti-phase coupling (alternation). The concentration of phase values, $r = \sqrt{(\text{average} \sin \varnothing)^2 + (\text{average} \cos \varnothing)^2}$, was determined as previously reported (*Kjaerulff and Kiehn, 1996*).

### Ladder walk and beam walk test

The animals were trained five times per session for 2 d before the testing day. At 4 wk after AAV-Syn-DIO-hM4Di-mCherry injections, the mice were subjected to the ladder walk and beam walk tests before and 30 min after CNO injections (5 mg/kg, i.p.).

The ladder walk test was used to assess skilled walking and limb placement (*Gerstmann et al., 2022*; *Metz and Whishaw, 2002*; *Ueno et al., 2012*). The horizontal ladder apparatus comprised side walls and metal rungs 13 cm above the ground. The metal rungs with 1.0 mm diameter were set at 2.0 cm intervals along a 1 m stretch. The walk through the 1 m stretch was recorded with a video camera (60 Hz, HDR-CX680, Sony), and the faulty placements of the fore- and hindpaw were counted. Complete miss and slipping off during weight-bearing were considered as faults (*Metz and Whishaw, 2002*). Each experiment was performed in triplicate, and the average number of faults was scored.

Beam walking test was used to assess motor coordination and balance while walking on the beam. The protocol was modified from the previous reports (*Gerstmann et al., 2022*; *Nakamura et al., 2011*). The plexiglass beam (1.0 cm in width, 50 cm in length) was suspended at 12 cm above the ground. The walking on the beam was recorded with a video camera, and the slips of the fore- and hindpaw were counted. The total number of slips in five runs per mouse were obtained.

### Statistical analysis

Quantitative data are represented as the mean ± standard error of the mean (SEM). Statistical analyses were performed by using Prism 6 (GraphPad). Pre- and post-injection groups were analyzed by the paired t-test and Wilcoxon matched-pairs signed rank test for parametric and nonparametric tests, respectively. A p-value less than 0.05 was considered statistically significant.

## Acknowledgements

We would like to thank C Zuker (Columbia University) for *Pkd2l1-Cre* mice; S Crone (CCHMC) and K Sharma (University of Chicago) for *Chx10*[Cre] mice; J Robbins (CCHMC) for CAG-lox-CAT-lox-EGFP mice; H Matsunami (Duke University) for anti-PKD2L1 antibody; S Tsuboguchi and K Ichikawa (Niigata University) for viral preparation; A Imai, N Hattori (National Institute for Physiological Sciences), and M Yatabe (Jichi Medical University) for their technical assistance on electron microscopy analyses; K Oda and T Sasaoka (Niigata University) for mouse rederivation; T Yamashita (Osaka University), K Shibuki, A Kakita, and O Onodera (Niigata University) for supporting materials. This work was supported by AMED-CREST (JP21gm1210005), Moonshot research (J21zf0127004), JSPS KAKENHI 17H04985, 17H05556, 17K19443, 20K21460, 21H02590, 21H05683, and Takeda Science Foundation (MU); JSPS KAKENHI JP16H06280 and JP22H04926, Grant-in-Aid for Scientific Research on Innovative Areas and Grant-in-Aid for Transformative Research Areas — Platforms for Advanced Technologies and Research Resources "Advanced Bioimaging Support", and 21H05241 (NO).

## Additional information

### Funding

| Funder | Grant reference number | Author |
| --- | --- | --- |
| AMED-CREST | JP21gm1210005 | Masaki Ueno |
| Moonshot research | J21zf0127004 | Masaki Ueno |

| Funder | Grant reference number | Author |
|---|---|---|
| JSPS KAKENHI | 17H04985 | Masaki Ueno |
| JSPS KAKENHI | 17H05556 | Masaki Ueno |
| JSPS KAKENHI | 17K19443 | Masaki Ueno |
| JSPS KAKENHI | 20K21460 | Masaki Ueno |
| JSPS KAKENHI | 21H02590 | Masaki Ueno |
| JSPS KAKENHI | 21H05683 | Masaki Ueno |
| Takeda Science Foundation | | Masaki Ueno |
| JSPS KAKENHI, Grant-in-Aid for Scientific Research on Innovative Areas - Platforms for Advanced Technologies and Research Resources "Advanced Bioimaging Support" | JP16H06280 | Nobuhiko Ohno |
| JSPS KAKENHI, Grant-in-Aid for Transformative Research Areas - Platforms for Advanced Technologies and Research Resources "Advanced Bioimaging Support" | JP22H04926 | Nobuhiko Ohno |
| JSPS KAKENHI | 21H05241 | Nobuhiko Ohno |

The funders had no role in study design, data collection and interpretation, or the decision to submit the work for publication.

## Author contributions

Yuka Nakamura, Conceptualization, Formal analysis, Investigation, Methodology, Writing – original draft; Miyuki Kurabe, Mami Matsumoto, Kana Hoshina, Investigation; Tokiharu Sato, Investigation, Methodology; Satoshi Miyashita, Data curation; Yoshinori Kamiya, Supervision; Kazuki Tainaka, Resources, Methodology; Hitoshi Matsuzawa, Software, Methodology; Nobuhiko Ohno, Supervision, Funding acquisition, Investigation, Methodology; Masaki Ueno, Conceptualization, Formal analysis, Supervision, Funding acquisition, Investigation, Methodology, Writing – original draft

## Author ORCIDs

Miyuki Kurabe http://orcid.org/0000-0003-1804-7037
Yoshinori Kamiya http://orcid.org/0000-0001-9790-9867
Masaki Ueno http://orcid.org/0000-0003-1484-9921

## Ethics

This study was performed in accordance with protocols approved by the Institutional Animal Care and Use Committee of Niigata University.

## Decision letter and Author response

Decision letter https://doi.org/10.7554/eLife.83108.sa1
Author response https://doi.org/10.7554/eLife.83108.sa2

# Additional files

## Supplementary files
• MDAR checklist

## Data availability
Figure 2 - Source Data 1 and Figure 7 - Source Data 1 contain the numerical data to used to generate the figures.

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
