## [Editor Report]

This study highlights a novel elegant approach to effectively target neurons contacting the cerebrospinal fluid in mice. The demonstration of the connectivity and roles of CSF-cNs to enhance locomotor speed is convincing and corroborates previous results obtained in mice and zebrafish. Overall this investigation is highly relevant for the sensorimotor and cerebrospinal fluid fields.

---

## [Decision Letter]

**Decision letter after peer review:**

Thank you for submitting your article "Cerebrospinal fluid-contacting neuron tracing reveals structural and functional connectivity for locomotion in the mouse spinal cord" for consideration by *eLife*. Your article has been reviewed by 3 peer reviewers, one of whom is a member of our Board of Reviewing Editors, and the evaluation has been overseen by Catherine Dulac as the Senior Editor. The following individuals involved in review of your submission have agreed to reveal their identity: Muriel Thoby-Brisson (Reviewer #2); Daniel Zytnicki (Reviewer #3).

The reviewers have appreciated the originality of your labeling approach and subsequent findings on the morphology and connectivity of sensory ciliated neurons contacting the cerebrospinal fluid. In a concerted manner, they request that you provide further details and evidence on the effect of inhibiting these cells on locomotion and on their local connectivity in the spinal cord.

Essential revisions:

Please address the requests below

1) Anatomical evidence:

a. Both the present study and that of Gerstmann (2022) indicate a functional link between CSF-cNs and motor skills. Please specify whether any higher arborization density was observed at spinal cord levels where locomotor CPG are known to reside. Did the authors observe any specificity in the CSF-cN anatomy in cervical or lumbar spinal cord?

b. A minor proportion of labeled CSF-cNs project bidirectionally (rostrally and caudally). Please specify where the somata of these cells reside. How does this observation fit with the general scheme proposed in Figure 7?

c. A poor terminal arborization is described for CSF-cNs with a cell body located at the lumbar level. Is this real or is it due to the difficulty of reconstructing a neuron over such a long distance? If this is real, could this morphology be linked to any different role played by these cells located in the lumbar spinal cord?

d. Is there any evidence for commissural connections between CSF-cNs in mice ?

2) Anatomical projections and optogenetic stimulation of CSF-cN neurons to infer synaptic connectivity:

a. The authors should provide more details on the procedure they followed. Where was the blue light applied? How many fibers are labelled/stimulated? Did the authors perform control experiments (no channelrhodopsin)? Specify other parameters of stimulation (intensity, location, duration). Did the authors test the GABAergic nature of the neurotransmitter involved (meaning repeating the stimulation in the presence of a specific blocker)?

b. The connectivity between CSF-cNs with axial motor neurons, V0c and V2a interneurons only relies on the anatomical study. Could the authors provide a more solid and full demonstration could be provided using again optogenetics tools and electrophysiological recordings in complement to the anatomical data? Is there some experimental limitation that prevents the authors to achieve such an investigation? In the absence of such data, the authors should acknowledge in the Discussion and in Figure 7 – supplement 1 that the connections between CSF-cNs, axial motor neurons, v0c and v2a interneurons are still hypothetical.

c. The concept of motor units has long been introduced by Sherrington to designate the ensemble made by a single motor neuron and the muscle fibers innervated by this motor neuron. This definition is widely accepted by the spinal cord physiologists all over the world. I strongly advice to keep this definition of motor units.

The concept of "motor unit" with the meaning suggested in the Discussion (p 32) is not fully established at this stage (there is no functional demonstration). Whatever the spinal connections are, the name "motor unit" is misleading and therefore MUST be changed (use "functional micromodule" instead).

3) Effects on locomotor function:

The size of the sample used to investigate locomotor parameters is rather low (at the limit to do trustable statistics). It remains unclear why the group of 7 individuals used to investigate the consequence on treadmill speed has not been entirely used for the other parameters measurements (step freq, stride, step). This part of the study is critical because it shows different results than the previous recent study published by Gerstmann et al. 2022. Consequently,

a. the authors must render their data fully convincing by increasing the number of animals analyzed in their assay. This would definitively strengthen their conclusions.

b. The authors should verify whether sensorimotor integration is also affected in the CSF-cN inhibited mouse on the ladder and rod assays as shown previously by Gerstmann et al. 2022.

*Reviewer #2 (Recommendations for the authors):*

The submitted paper focuses on cerebrospinal fluid-contacting neurons (CSF-cNs) that are mechano-and chemosensors located near the central canal of the spinal cord, and provides a new specific method to label and manipulate mouse CSF-cNs. This study revealed CSF-cNs structures, connectivity and involvement in locomotor function. By combining viral tracing, anatomical reconstruction, electronic microscopy, otpogenetic stimulation and in vivo analysis of locomotion the authors describe CSF-cNs rostral axon extension over a large distance (2000-4000µm), demonstrate their recurrent interconnection and their connection with different neuronal elements of the spinal locomotor circuits. The present study is an excellent complement to a previous study published recently (Gerstmann et al., 2022). Experiments are well performed, figures and findings are of very high quality and the data are strongly supporting the conclusions. Nevertheless, the authors should address a few points detailed below.

1) Anatomical data:

– Both the present study and that of Gerstmann (2022) indicate a functional link between CSF-cNs and motor skills. I was wondering whether any higher arborization density was observed at spinal cord levels where locomotor CPG are known to reside. Did the authors observe any specific anatomy in these spinal regions?

– A minor proportion of labeled CSF-cNs seem to project bidirectionally (rostrally and caudally). Are they found along the entire spinal cord length or are they preferentially located at the cervical level as shown in Figure 2? How does this fit with the general scheme proposed in Figure 7?

– A poor terminal arborization is described for CSF-cNs with cell body located at the lumbar level. Is this a trustable feature or is it due to the difficulty of reconstructing a neuron over such a long distance? Could this be linked to any different role played by these cells located in the lumbar spinal cord?

– Is there any evidence for commissural connections between CSF-cNs?

2) Optogenetic stimulation of CSF-cN neurons to infer synaptic connectivity:

The authors should provide more details in the procedure followed. For instance where is the blue light applied? How many fibers are labelled/stimulated? Have control experiments been performed (no channelrhodopsin? Other parameters of stimulation intensity, location, length)? Did the authors test the GABAergic nature of the neurotransmitter involved (meaning repeating the stimulation in the presence of a specific blocker)?

3) Locomotor function: The size of the sample used to investigate locomotor parameters is rather low (at the limit to do trustable statistics). It remains unclear why the group of 7 individuals used to investigate the consequence on treadmill speed has not been entirely used for the other parameters measurements (step freq, stride, step). This part of the study is important because it shows results that are different/complementary (but not necessarily contradictory) to a previous recent study published by Gerstmann (2022). So for this specific point the authors must render their data fully convincing by increasing the number of animals analyzed. This would definitively strengthen their conclusions.

4) The authors explain that to specifically label a small proportion of CSF-cNs they injected a low dose of AAV-Syn-Cre intraventricularly (line 245) and later (line 273) they injected a higher dose. Details must be given for these different doses.

*Reviewer #3 (Recommendations for the authors):*

The connectivity between CSF-cNs with axial motor neurons, V0c and V2a interneurons only relies on the anatomical study. We may wonder whether a more solid and full demonstration could be provided using again optogenetics tools and electrophysiological recordings in complement to the anatomical data? Is there some experimental limitation that prevents the authors to achieve such a study? In the absence of such data, the authors should acknowledge in the Discussion and in Figure 7 – supplement 1 that the connections between CSF-cNs, axial motor neurons, v0c and v2a interneurons are still hypothetical.

The concept of "motor unit" with the meaning suggested in the Discussion (p 32) is not fully established at this stage (there is no functional demonstration).

Whatever the spinal connections are, I strongly emphasize that the name "motor unit" is really misleading and MUST be changed (maybe "functional micromodule" would be more appropriate?). Indeed, the concept of motor units has long been introduced by Sherrington to designate the ensemble made by a single motor neuron and the muscle fibers innervated by this motor neuron. This definition is widely accepted by spinal cord physiologists all over the world. I strongly advise keeping this definition of motor units.

---

## [Author Response]

Essential revisions:Please address the requests below1) Anatomical evidence:a. Both the present study and that of Gerstmann (2022) indicate a functional link between CSF-cNs and motor skills. Please specify whether any higher arborization density was observed at spinal cord levels where locomotor CPG are known to reside. Did the authors observe any specificity in the CSF-cN anatomy in cervical or lumbar spinal cord?

Thank you for critical and important comments on the anatomical analyses. It remains unclear whether CSF-cNs outputs are biased to specific spinal levels containing locomotor CPG. The density of dorsally innervating collaterals was higher at the cervical levels and gradually decreased to the caudal spinal cord (Figure 1–supplement 1I–L, Figure 2I, Figure 2–supplement 1). This suggests that the dorsal innervations might not correlate with the spinal levels of CPGs located at C5–T1 and T12–L5 (Kjaerulff et al., 1996; Gordon et al., 2008 etc.). We also note that CSF-cNs had connections within the ventral bundles that existed throughout the spinal levels. This further suggests that CSF-cNs may not function specifically in CPG regions. As CSF-cNs were present throughout the spinal cord, we assume that they influence the circuitry such as for axial muscles at various spinal levels. However, more sophisticated investigations are required to understand the differences or similarity of the functions among spinal levels. We added these points in Results and Discussion.

b. A minor proportion of labeled CSF-cNs project bidirectionally (rostrally and caudally). Please specify where the somata of these cells reside. How does this observation fit with the general scheme proposed in Figure 7?

The bidirectionally-projecting CSF-cNs were located only at the cervical level C2 (two cells) and C4/5 (two cells), in which at the ventral (three cells) and lateral (one cell) side of the central canal, in our preparation. We do not know if they are classified as minor subtype or mis-projection CSF-cNs. They were rarely observed only at the cervical level (four cells in total 332), and thus it was difficult to fit and integrate them in the model of Figure 7-supplement 1, at present. It requires further studies to examine their features. We added these points in Results, Discussion, and Figure legends of Figure 7-supplement 1.

c. A poor terminal arborization is described for CSF-cNs with a cell body located at the lumbar level. Is this real or is it due to the difficulty of reconstructing a neuron over such a long distance? If this is real, could this morphology be linked to any different role played by these cells located in the lumbar spinal cord?

We showed the data of traced CSF-cNs; some axons clearly ended in the ventral funiculus and did not have many dorsal collaterals especially at the lumbar and sacral levels. This is also suggested by the observation of cleared spinal cord, exhibiting less amount of GFP^+^ fiber bundles at the ventral midline especially at the lumbar and sacral level (Figure 1–supplement 1I–L), although we could not clearly discriminate dorsal collaterals from ventrally-projecting axons there.

Despite the less dorsal collaterals, we observed synaptic structures within the ventral funiculus, in which neurites received multiple inputs from CSF-cNs with a typical rosette-like structure. Thus, CSF-cNs outputs would also function in the caudal spinal cord. The structural difference may suggest functional difference among the spinal levels, but unfortunately we do not have clear answers to explain it at present. We state this issue in Discussion.

d. Is there any evidence for commissural connections between CSF-cNs in mice ?

We additionally analyzed the projection side of single-cell traced CSF-cNs in three structural parts; (1) ventral projections to the ventral funiculus, (2) rostral projections in the ventral funiculus, and (3) dorsal projections to the central canal.

As for the (1) ventral projections, since most of the labeled CSF-cNs soma were located at the ventral midline along the central canal (67/71 cells), it was difficult to strictly determine if they projected to ipsi- or contralateral side. However, all the axons projected to specific side without bilateral projections (left side, 46.5%; right side, 53.5%). The remaining four dorsolateral cells projected axons ipsilaterally. In the (2) rostral projections in the ventral funiculus, we did not see any fibers crossing the midline. Lastly, at the (3) dorsal projections back to the central canal, in many cases, the fibers extended not only to the ipsilateral but also to the contralateral side (ipsilateral, 7.0%; bidirectional, 62.8%; contralateral, 9.3%; midline beneath the central canal, 20.9%). It requires further examinations if they had functional connections on either side. Moreover, the possibly conserved ipsilateral projections should be examined in the dorsolateral CSF-cNs of the central canal, which could not be effectively analyzed in the present study. We added these points in Results, Discussion, and Materials and methods.

2) Anatomical projections and optogenetic stimulation of CSF-cN neurons to infer synaptic connectivity:a. The authors should provide more details on the procedure they followed. Where was the blue light applied? How many fibers are labelled/stimulated?

We described more detail procedures in Materials and methods: “To elicit evoked IPSCs, photostimulation with a 470-nm blue light was used with an LED driver (LEDD1B – T-Cube LED driver, Mounted LED for microscope, Thorlabs), which was passed through a filter (LED-DA/FI/TX-A-000, Optline) and an immersion objective (40x/0.8 W, Nikon) directly above the spinal cord slice (distance, less than 3 mm). The stimuli (duration, 10 ms; frequency, 0.3 Hz; repetition, 10 times; light intensity, ~5 mW/mm^2^) were controlled with an STO mk II stimulator (BRC). GABAA receptor-evoked IPSCs were isolated by bath perfusion of 20 µM Bicuculline (Sigma).”. We also counted the number of ChR2-mCherry-labeled fibers; 355.3 ± 52.5 fibers (n = 3) were observed at T4–T6 levels, indicating the efficiency of the labeling. We added these points in Results and Materials and methods.

Did the authors perform control experiments (no channelrhodopsin)? Specify other parameters of stimulation (intensity, location, duration). Did the authors test the GABAergic nature of the neurotransmitter involved (meaning repeating the stimulation in the presence of a specific blocker)?

We performed control experiments in spinal slices of *Pkd2l1*-*Cre*;*lCl^-^GFP* mice injected with AAV-Syn-DIO-mCherry. No responses were detected by a blue light stimulation (0/14 cells). We added the new data in Figure 5D and Results. We further performed recordings under treatment with bicuculline, a GABAA receptor antagonist. It blocked the IPSC responses of CSF-cNs (8/8 cells), indicating that the IPSCs were mediated by GABAA receptors. We added these new data in Figure 5C and Results.

b. The connectivity between CSF-cNs with axial motor neurons, V0c and V2a interneurons only relies on the anatomical study. Could the authors provide a more solid and full demonstration could be provided using again optogenetics tools and electrophysiological recordings in complement to the anatomical data? Is there some experimental limitation that prevents the authors to achieve such an investigation? In the absence of such data, the authors should acknowledge in the Discussion and in Figure 7 – supplement 1 that the connections between CSF-cNs, axial motor neurons, v0c and v2a interneurons are still hypothetical.

We agree with the reviewer’s comment that electrophysiological recordings are required to determine their connectivity. However, recordings from V2a or V0c neurons were rather challenging for us at present. Triple transgenic mice such as *Chx10* (*or Chat*)-*Cre*;*Pkd2l1*-*Cre*;*lox*-*CAT*-*lox*-*GFP* mice may be available to identify the cells; however, it will be difficult to avoid leaking ChR2 expression in V2a and V0c neurons by AAV-Syn-DIO-ChR2 injections. A simple method to fluorescently label those neurons such as by developing *Chx10* (*or Chat*)-*GFP* mice and crossing with *Pkd2l1*-*Cre* mice will be required for further analyses.

The recordings from motor neurons in the medial column were also difficult in our technique. In particular, it was difficult to stably record motor neurons at whole cell configuration even in some rare cases of success in patch clamp around P28. The recent paper by Gerstmann et al. (*Curr Biol* 2022) also described that they could not detect evoked IPSCs in the motor neurons, in which they discussed that the discrepancy with the anatomical findings could be attributed to sparse connectivity and technical limitation to find intact connections in transverse slices. Therefore, an alternative methodology would be needed to determine their connections. As functional connections with spinal interneurons and motor neurons were difficult to assess in our study, we revised the Discussion and Figure legends of Figure 7–supplement 1 to state that the connections and the proposed model are still hypothetical.

c. The concept of motor units has long been introduced by Sherrington to designate the ensemble made by a single motor neuron and the muscle fibers innervated by this motor neuron. This definition is widely accepted by the spinal cord physiologists all over the world. I strongly advice to keep this definition of motor units.The concept of "motor unit" with the meaning suggested in the Discussion (p 32) is not fully established at this stage (there is no functional demonstration). Whatever the spinal connections are, the name "motor unit" is misleading and therefore MUST be changed (use "functional micromodule" instead).

We appreciate this critical comment. Considering the comment and the definition of Sherrington’s and others on motor neurons, we replaced the word to “module” throughout the manuscript (e.g. Kiehn, *Nat Rev Neurosci* 2016; Caggiano et al., *Sci Rep* 2016; Grillner et al., *Physiol Rev* 2020).

3) Effects on locomotor function:The size of the sample used to investigate locomotor parameters is rather low (at the limit to do trustable statistics). It remains unclear why the group of 7 individuals used to investigate the consequence on treadmill speed has not been entirely used for the other parameters measurements (step freq, stride, step). This part of the study is critical because it shows different results than the previous recent study published by Gerstmann et al. 2022. Consequently,a. the authors must render their data fully convincing by increasing the number of animals analyzed in their assay. This would definitively strengthen their conclusions.

We increased the number of animals (total n = 7) for the analyses of step frequency, stride, and step lengths. The data showed consistent tendency of reduced step frequency and increased stride length in *Pkd2l1*-*Cre* group. We added these new data in Figure 7C–K.

b. The authors should verify whether sensorimotor integration is also affected in the CSF-cN inhibited mouse on the ladder and rod assays as shown previously by Gerstmann et al. 2022.

We additionally performed ladder walk and beam walk tests in WT and *Pkd2l1*-*Cre* mice injected with AAV-Syn-DIO-hM4Di and CNO. The scores were worsened in the ladder walk test, while the beam walk test did not show significant differences. The new data were added in Results, Figure 7L–O, Discussion, and Materials and methods. The absence of phenotype in our beam walk test might be attributed to differences in methodology or sensitivity (chemogenetics, ventral CSF-cNs targeting in our study vs. diphtheria toxin, whole CSF-cNs; different apparatus etc.) from that in the study by Gerstmann et al. 2022.